# Multi-species atlas resolves an axolotl limb development and regeneration paradox

Jixing Zhong[1,4], Rita Aires [2,4], Georgios Tsissios[1], Evangelia Skoufa[1], Kerstin Brandt[3], Tatiana Sandoval-Guzmán [2,3] ✉ & Can Aztekin [1] ✉

Humans and other tetrapods are considered to require apical-ectodermal-ridge (AER) cells for limb development, and AER-like cells are suggested to be re-formed to initiate limb regeneration. Paradoxically, the presence of AER in the axolotl, a primary model organism for regeneration, remains controversial. Here, by leveraging a single-cell transcriptomics-based multi-species atlas, composed of axolotl, human, mouse, chicken, and frog cells, we first establish that axolotls contain cells with AER characteristics. Further analyses and spatial transcriptomics reveal that axolotl limbs do not fully re-form AER cells during regeneration. Moreover, the axolotl mesoderm displays part of the AER machinery, revealing a program for limb (re)growth. These results clarify the debate about the axolotl AER and the extent to which the limb developmental program is recapitulated during regeneration.

Vertebrate limb development requires apical-ectodermal-ridge, AER, cells at the dorsal-ventral boundary of developing limbs, which enable the expansion of limb bud mesodermal cells and provide patterning cues[1]. The AER supplies critical signaling ligands and its spatial organization contributes to the morphogen gradients to form a correctly patterned limb[2]. Interestingly, the leading limb regeneration model organism, the axolotl[3], was suggested not to have an AER, as they lack a morphological ridge structure and some of the molecular AER markers[4–7]. Despite these findings, as regeneration is thought to largely recapitulate development, salamander regeneration is considered to re-form AER-like cells to act as a signaling center apical-epithelial-cap, AEC, at the amputation plane[8–10]. The AEC is required to form a connective tissue lineage-rich blastema for regeneration[11,12], and its absence in mammals has been long-hypothesized to be one of the reasons for the regeneration-incompetency[13,14]. However, previous reports provided conflicting results for the AER and AEC marker expressions[4,7,15–21], and a sub-group of the axolotl connective tissue (CT) cells was suggested to express some of the AER-related genes[4,7,21,22]. Consequently, different conclusions have been drawn based either on morphological assessments or assaying the expression of a small set of specific marker genes. Because of this, the existence of AER in axolotls and the re-use of AER-like cells for salamander limb regeneration remain unclear. Unbiased and comprehensive analyses are warranted to resolve the cellular identity of this critical population in the contexts of limb development, regeneration, and evolution studies.

Here, we establish a single-cell transcriptomics-based multi-species limb atlas for five vertebrates, including the axolotl. Using the atlas and imaging, we show that the developing axolotl limb bud contains cells with AER characteristics. By comparing limb development and regeneration using single-cell transcriptomics and spatial transcriptomics on regenerating limbs, we show that the axolotl AER program is not fully recapitulated during regeneration. Furthermore, we demonstrate that the regenerating mesoderm displays a subset of the AER machinery, and that this is an axolotl-specific feature. These results not only provide a comprehensive assessment of the differences and similarities between limb development across species, but also reveal that different cell types are present during limb development and regeneration in axolotls.

[1]School of Life Sciences, Swiss Federal Institute of Technology Lausanne, EPFL, 1015 Lausanne, Switzerland. [2]Department of Internal Medicine III, Center for Healthy Aging, University Hospital Carl Gustav Carus, Technische Universität Dresden, Dresden, Germany. [3]Paul Langerhans Institute Dresden, Helmholtz Centre Munich, University Hospital Carl Gustav Carus, Technische Universität Dresden, Dresden, Germany. [4]These authors contributed equally: Jixing Zhong, Rita Aires. ✉e-mail: tatiana.sandoval_guzman@tu-dresden.de; can.aztekin@epfl.ch

## Results

### Multi-species limb atlas of five vertebrates

To determine if the developing axolotl limbs have an AER-like population, we aimed to benefit from a single-cell transcriptomics-based multi-species limb atlas (Fig. 1a). We hypothesized that if AER-like cells are present in the axolotl limb, their molecular identity would overlap with the AER cells from other species. To test this, we first collected publicly available scRNA-Seq datasets of developing limbs from axolotls (*Ambystoma mexicanum*)[23] and, species with their developmental stages that are documented to have an AER: humans (*Homo sapiens*)[24,25] mice (*Mus musculus*)[26,27], chickens (*Gallus gallus*)[28], and frogs (*Xenopus laevis*)[23,29]) (Supplementary Figs. 1 and 2 and Supplementary Data 1). Then, using Seurat integration, which has been shown to integrate cross-species datasets with high accuracy[30], we established a multi-species limb atlas that in total contains 50,248 representative cells from all samples (Fig. 1b and Supplementary Fig. 3a–e). Coarse lineage annotation detected various mesodermal and ectodermal populations in the multi-species limb atlas, and finer annotation captured the AER cluster (Fig. 1c and Supplementary Fig. 3a–e). Critically, the goblet cell cluster, which is prevalent in amphibian skin, was dominated by frog and axolotl cells, with minimal representation (~2.3% in total) of mouse, human, and chicken cells (Supplementary Fig. 3f), emphasizing that the established multi-species atlas preserves species-specific variances. Furthermore, when we integrated an E16.5 stylopod dataset, which should contain no AER cells, as a negative control, we detected no contribution to the AER cluster from this dataset (Supplementary Fig. 3g), suggesting that our approach provided robust integration with no detected over-correction issue. Overall, the multi-species atlas was able to group cells based on their potential cell identity rather than the species or other technical factors (Supplementary Fig. 3a–c).

### Axolotl limb buds contain cells with AER features

Examining the species contribution to the multi-species AER cluster, we found cells from developing axolotl limbs, suggesting that axolotl limb buds contain AER-like cells (Fig. 1d). A second cross-species data integration approach, SAMap[31], yielded the same result, suggesting that axolotl limb buds contain AER-like cells regardless of the integration method (Supplementary Fig. 4). Moreover, the presence of axolotl cells in the multi-species AER cluster was not affected by the variation in sequencing quality among datasets, as the omission of the chicken dataset (which is of lower quality than other datasets) from the atlas did not change the results (Supplementary Fig. 3h).

Then, we surveyed the established AER markers to determine the transcriptional similarities of these axolotl cells with the AER in other species (Fig. 1e and Supplementary Fig. 5). The large majority of axolotl cells in this cluster (~97% expressing at least 3 of the 18 listed AER markers and 85% expressing at least 5) co-expressed many of the AER markers[1], such as *Wnt5a* and *Msx2*, whilst some others (e.g., *Fgf8*) were absent (Fig. 1e and Supplementary Fig. 5), in alignment with recent reports[4,7]. A further transcriptome-wide comparison found not only a high similarity of AER-like axolotl cells to the AER cells in other species, but also that this population was distinct from non-AER basal ectodermal cells (Fig. 1f and Supplementary Fig. 6). Hence, our results provide high-throughput evidence that developing axolotl limbs contain cells with AER transcriptional programs as in other species, and these cells are hereafter referred to as axolotl AER cells.

Next, we evaluated the potential functional properties of the axolotl AER. As the AER is a well-recognized signaling center[1], we aimed to reveal if axolotl AER cells express ligands from critical limb development-related signaling pathways. We generated a potential secretome gene group, composed of ligands from FGFs, BMPs, WNTs, NOTCH/DELTAs, and TGFbs (Supplementary Data 2), and performed gene set enrichment analysis on cell clusters. Axolotl AER cells had comparable signaling potential to the AER in other species (Fig. 1g and Supplementary Fig. 7). Meanwhile, individual pathway examination revealed the axolotl AER differs in paralog and co-factor expressions, particularly in the FGF and WNT pathways (Supplementary Fig. 8), complementing and extending previous observations[4,7,21]. We could not detect noticeable differences in transcriptional levels associated with DELTAs, BMPs, or TGFbs (Supplementary Fig. 8). Overall, axolotl AER cells have comparable transcriptional levels for limb development-related signaling pathway ligands, albeit potentially critical differences remain.

The AER in other species forms at the dorsal-ventral boundary of developing limbs, which is considered to be critical in setting morphogen gradients for subsequent growth and patterning[2]. To determine if the axolotl AER cells have a similar spatial organization, we sought to visualize them. Using the scRNA-seq dataset, we identified *Dr999-Pmt21178* and *Vwa2* as the marker genes with high expression specifically in the axolotl AER cells, although these two genes also had weak expression in the non-AER basal ectoderm (Supplementary Fig. 9a–c). We then performed the whole-mount hybridization chain reaction (HCR), which is a semi-quantitative mRNA visualization method, of these two marker genes on developing axolotl limbs from different stages. *Dr999-Pmt21178* and *Vwa2* showed specific expression at the dorsal-ventral boundary at the limb bud stages, albeit more scattered compared to AER localization in other species, (Figs. 1h, 2a, and Supplementary Fig. 9d), and digit tips during digit forming stages (Figs. 1i and 2a), resembling, but not identical to, the spatial organization of AER in other species[1,2]. We found that *Dr999-Pmt21178* expression colocalized with the pan-ectoderm marker *Epcam* only in the distal limb bud ectoderm (Fig. 2b), suggesting that the axolotl AER-like cells are distinct from non-AER ectoderm. Moreover, we then showed that *Dr999-Pmt21178* positive cells also co-express a known AER marker *Msx2* (Fig. 2c, d).

Since our results revealed *Vwa2* and *Dr999-Pmt21178* as markers for the AER-like cells in axolotls, we then examined their expression in the AER in other species. Unlike *Dr999-Pmt21178* which does not have orthologs in other analyzed species, *Vwa2* is expressed at different levels in the AER of humans, mice, and frogs in scRNA-seq data (Supplementary Fig 10a). Meanwhile, we did not detect any *Vwa2*-expressing cells in the chicken dataset, which may be due to its low sequencing depth (Supplementary Figs. 1 and 10a). Indeed, performing HCR on developing chicken or mouse limb buds confirmed *Vwa2* expression in mouse and chicken AERs (Supplementary Figs. 10b–e). Thus, these results indicate that employing cross-species comparisons can unveil novel cell-type markers.

To explore species-specific features, we focused on the expressions of FGF ligands in the axolotl AER-like cells as they are well associated with the AER functions. We found that the FGF ligand expression profile showed variability across species, even between humans and mice (Supplementary Fig. 8). Specifically, scRNA-seq suggested that a subset of axolotl AER-like cells show specific expression of *Fgf7*, *Fgf16*, and *Fgf18*, although they did not express mouse AER-FGFs (*Fgf4*, *Fgf8*, *Fgf9*, *Fgf17*) (Supplementary Fig. 8). When we performed HCR against *Fgf7* and *Fgf18*, we found that they are predominantly expressed at the dorsal-ventral boundary of the distal limb bud ectoderm (Supplementary Fig. 11), similar to the mouse AER FGFs. Taken together, these results indicate a diversification of FGF ligands in the axolotl ectoderm.

Next, we investigated the AER cellular morphology. Unlike amniotic or frog AERs where AER cells have mainly cuboidal or columnar cell shape[1,29], we found that *Dr999-Pmt21178* positive axolotl AER cells mostly present a squamous shape (Figs. 1j and 2a). Moreover, these AER cells have a high degree of similarity to the outer skin cells, the periderm (Figs. 1j and 2a), which may explain why prior morphology-based studies could not distinguish them from other populations. Altogether, our imaging results highlight that the axolotl

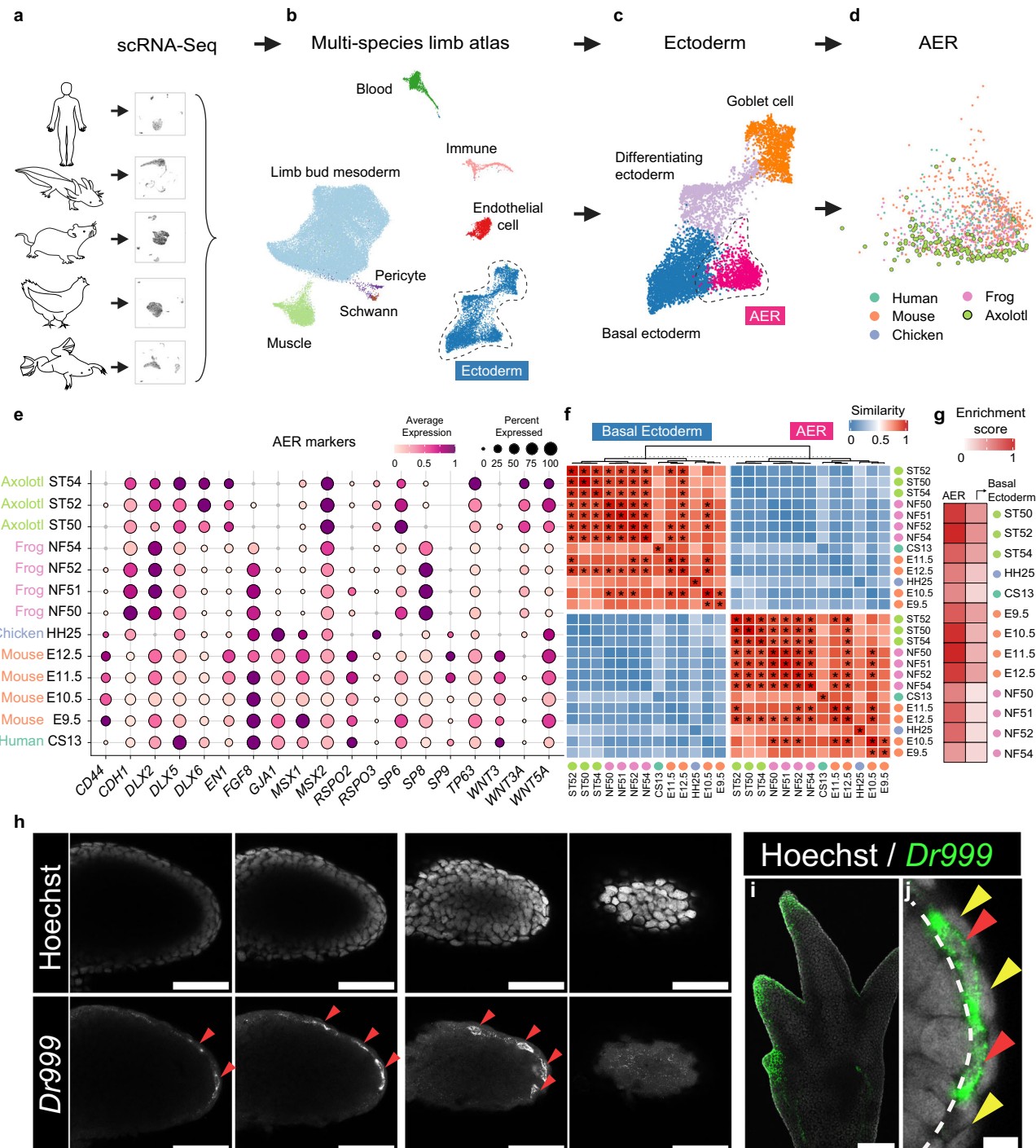

**Fig. 1 | Multi-species limb atlas reveals developing axolotl limbs have cells with apical-ectodermal-ridge (AER) characteristics. a** Schematic to generate a multi-species limb atlas using publicly available single-cell RNA-Seq (scRNA-Seq) datasets. **b** UMAP plot of Seurat-integrated multi-species limb atlas. Individual datasets from each species with different developmental stages are integrated. Dots are colored by cell identities. **c** UMAP plot of the ectodermal lineage. Dots are colored by cell identities. **d** UMAP plot of species contribution to the AER cluster. Dots are colored by species. **e** Dot plot showing AER marker expressions in AER cells from different species. The dot color indicates the mean expression that was normalized to the max of each dataset and to the max of each gene; the dot size represents the percentage of cells with non-zero expression. **f** Heatmap showing the Meta-Neighbor score for pair-wise similarities of basal ectoderm and AER clusters. X- and Y-axes indicate species and developmental stages. Asterisks (*) denote the pairs with scores above 0.9. Source data provided as a Source data file. **g** Heatmap showing signaling ligands gene set enrichment analysis scores for AER, and non-

AER basal ectoderm clusters. The basal ectoderm represents the transcriptome-wide most similar population to the AER, and is used for comparison. Colored dots in the Y-axis indicate different species. Source data provided as a Source data file. **h** Single optical section of z-stacks of confocal images of Stage 46 axolotl forelimb buds stained for *Dr999-Pmt21178* (referred to as *Dr999*) mRNA via hybridization-chain-reaction (HCR). Different z-stacks were shown from left to right, representing different levels of the dorsal-ventral axis. (Top) Gray, Hoechst; Bottom Gray, *Dr999* mRNA. Scale bar: 100 μm. **i** Max-projection confocal image of Stage 53 axolotl forelimb digit tips stained for *Dr999* mRNA via HCR. Green, *Dr999* mRNA; Gray, Hoechst. Scale bar: 250 μm. **j** Zoomed single optical section image of the axolotl limb bud from (h) stained for *Dr999* mRNA. Red arrows show *Dr999*+ squamous cells, and yellow arrows show outer layer peridermal cells. The basement membrane is labeled with a dashed line. Green, *Dr999* mRNA; Gray, Hoechst. Scale bar: 10 μm.

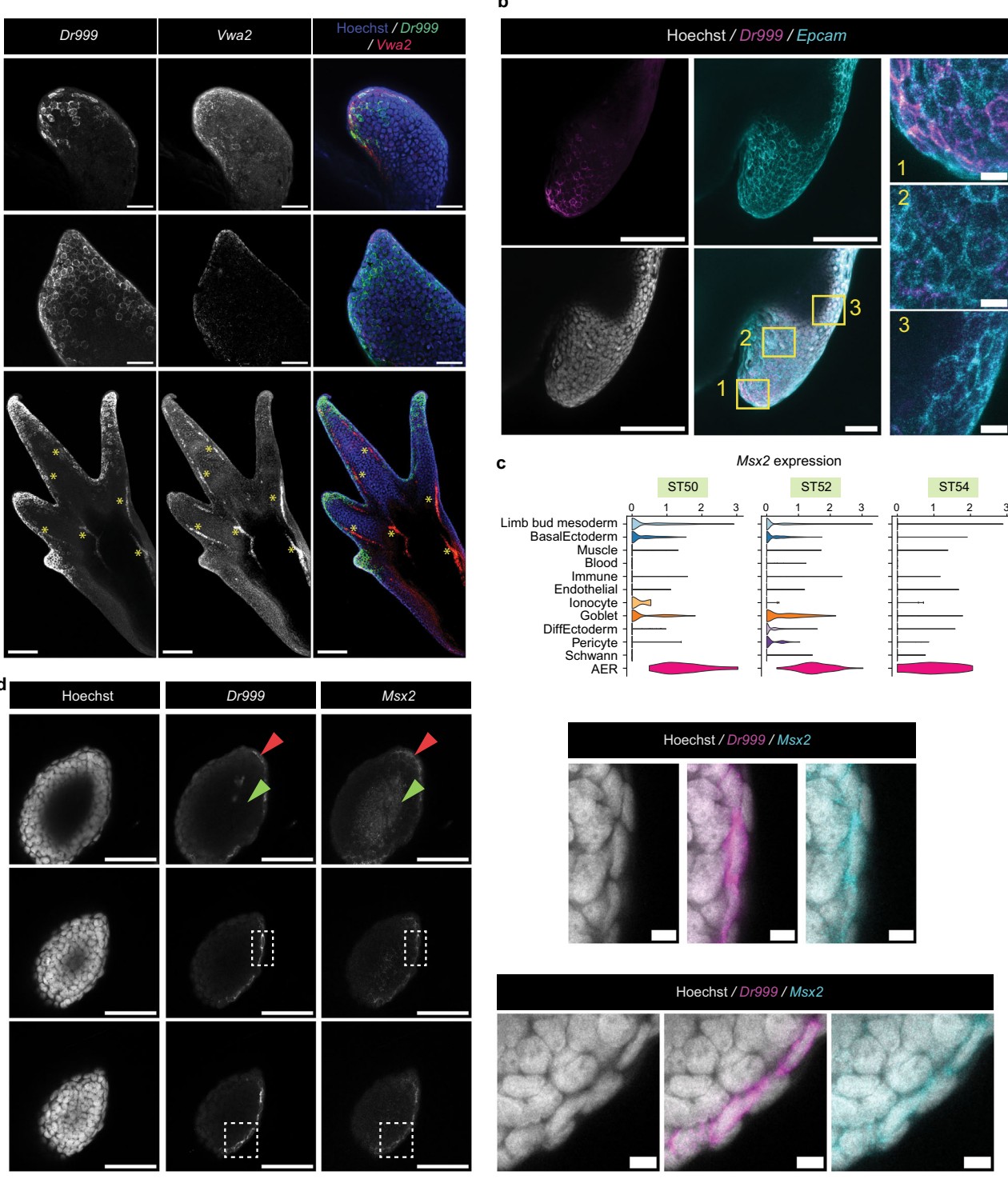

**Fig. 2 | Spatial organization of axolotl AER cells during limb development.**
**a** Max-projection confocal images of ST 46 (top), 50 (mid), and 53 (bottom) axolotl forelimbs correspond to ST50, 52, 54 hindlimb scRNA-Seq data, respectively, stained for *Dr999* and *Vwa2* mRNA via HCR. Green, *Dr999* mRNA; Red, *Vwa2* mRNA; Blue, Hoechst. Yellow * is added to the bottom images to indicate auto-fluorescence. Scale bar: 100 μm (top and mid) and 200 μm (bottom). **b** Max-projection confocal image of axolotl ST 53 hindlimb buds stained for *Dr999* and *Epcam* mRNA via HCR. Gray, Hoechst; Magenta, *Dr999* mRNA; Cyan, *Epcam* mRNA. Enlarged views of corresponding regions indicated by the numbered yellow boxes

are shown side by side. Scale bar: 20 μm for zoomed images, 100 μm for the merged image and 200 μm for the others. **c** *Msx2* expression in axolotl limb buds based on the scRNA-Seq datasets. **d** (Left) Max-projection confocal image of axolotl Stage 53 hindlimb buds stained for *Dr999* and *Msx2* mRNA via HCR. Scale bar: 100 μm. Red arrows show skin cells, and the green arrows show non-skin meso-dermal cells to highlight *Msx2* staining profile. (Right) Close-up images of the dashed labeled region of *Dr999* and *Msx2* double-positive skin cells. Scale bar: 10 μm.

AER cells have a spatial organization in developing limbs similar to other species with a unique cellular morphology.

### Axolotl AER is not fully re-formed during limb regeneration

Earlier studies suggested that the AER is re-formed during limb regeneration across different species acting as the AEC, based on morphological examination and a set of gene expression similarities with other species[8,9]. In alignment with this proposition, previously, we found that at the single-cell level, limb regeneration-competent *Xenopus laevis* tadpoles re-use AER cell transcriptional program to act as the AEC[29] (Supplementary Fig. 12). Having identified axolotl limb buds containing cells with an AER transcriptional program, we asked if this program is re-used in the course of axolotl regeneration, as in frog tadpole regeneration.

To test whether the axolotl AER program is re-used during regeneration, we first combined publicly available comprehensive and time-course axolotl limb regeneration (Supplementary Fig. 13)[19] and development (Supplementary Fig. 2e)[23] datasets (Fig. 3a). We then focused on all the basal ectodermal cells, which would contain the AER or AEC cells. Subclustering of these basal ectodermal cells revealed that the axolotl AER cells gathered with some cells from the regeneration samples (Fig. 3b, c). Investigation of the expression profile of these cells from the regeneration samples revealed highly specific expression of the known axolotl AEC markers (e.g., *Mdk*[12], *Frem2*[18], *Krt5*[32]) (Supplementary Fig. 13d), indicating significant similarity between the AER and the AEC. Nevertheless, we found that these AER and AEC cells have different localizations within the same cluster (Fig. 3c), highlighting the potential transcriptional differences between them. Indeed, the comparison of AER and AEC cells by differentially expressed gene analysis identified significant transcriptional changes (Fig. 3d, and Supplementary Fig. 14 and Supplementary Data 3), some of which are specific to post-amputation samples (e.g., *Mmp13*) (Fig. 3d, and Supplementary Fig. 14).

Then, we asked if we could detect comparable signaling properties in the axolotl AEC to the axolotl AER cells. However, we failed to identify high levels of expression for some of the ligands belonging to the mainly studied signaling pathways in this population, which contrasted with their developmental counterparts (Fig. 3e). Specifically, in axolotl AEC cells there was a lack of FGF pathway related ligands and quantitative expression differences for other ligands (Fig. 3e).

To validate these results and eliminate the possibility that scRNA-Seq did not capture cells specifically from the amputation plane, we performed spatial transcriptomics on a regenerating limb where the AEC and the blastema were evident (Fig. 3f–h and Supplementary Fig. 15). With this data we could identify the spatial distribution of the expected tissues types, including the AEC cluster located at the tip of the amputated limbs, as well as the blastema (Fig. 3g, h and Supplementary Fig. 15), both of which were previously difficult to pinpoint using the conventional scRNA-Seq approach in axolotls. Nonetheless, when evaluating this AEC cluster, we could not detect certain AER marker genes and high ligand expressions again (Fig. 3d, e). Altogether, our results suggest the axolotl AEC is distinct from the axolotl AER developmental program. Critically, it lacks a high level of expression for developmental signaling ligands, unlike in frog tadpoles, emphasizing different regenerative programs between these two species.

### Axolotl mesoderm exhibits AER program during regeneration

Interestingly, a few of the AER-associated genes, such as *Fgf8*, were reported to be expressed in the salamander anterior mesoderm[4,7,22], raising the possibility that axolotl mesodermal lineage might show features of the signaling center epithelial AER transcriptional program. Beyond single-gene investigations, we sought to comprehensively evaluate this possibility.

To test this, we leveraged our multi-species limb atlas, which underscored a transcriptional program related to AER cell identity across analyzed species (Fig. 1c, d; Supplementary Figs. 3 and 4). First, we identified the differentially expressed genes in the AER cluster in the atlas (Supplementary Data 4). Second, we used consensus non-negative matrix factorization (cNMF) to detect transcriptional modules related to cell-identity and cell-activity within specific populations[33] (Supplementary Fig. 16 and Supplementary Data 4). Then, using these AER-related gene sets, we surveyed the expression pattern in mesodermal populations, including using gene set enrichment analysis (Supplementary Fig. 17). In parallel, we also tested whether clustering based on these gene sets would aggregate AER-related ectodermal cells and mesodermal cells together, indicative of a high degree of similarity (Fig. 4a–c and Supplementary Figs. 18–22).

Remarkably, our approach identified AEC cells and a subset of axolotl CT cells to have high enrichment scores for the differentially expressed genes in the AER cluster (Supplementary Fig. 17b) and gather together in the axolotl regeneration datasets (Fig. 4c and Supplementary Figs. 18–22). Subsequently, we found that the shared genes between these grouped CT cells and AEC include certain epithelial AER genes (e.g., *Jag2*, *Cdh1*), although some of the previously reported genes, such as *Fgf8*[7], were not detected in this population (Fig. 4d), which might be due to the limitations to the analyzed dataset (e.g. timing of sample collection, enrichment of non-anterior mesodermal lineage cells). Moreover, we identified these CT cells express AER-associated signaling ligands (e.g., *Bambi*, *Bmp2*) (Supplementary Fig. 23a), as well as other genes such as *Vwde*, *Mdk*, and *Krt18* (Supplementary Fig. 23b) that have been already demonstrated to be critical for successful salamander limb regeneration and blastema proliferation[12,22,34].

Then, using our spatial transcriptomics data, we confirmed the presence of the CT cells showing part of the AER program and revealed that they are mostly present in the blastema, where they express both fibroblast- and epithelial-associated genes (Fig. 4e–g), which was also in alignment with the reported mesodermal *Krt5*, *Krt17*, and *Krt18* expression[22,35,36]. Moreover, we found that some CT cells showing the AER program are also present in the intact axolotl limbs before limb amputations (Supplementary Fig. 23c). Importantly, in limb development datasets of species analyzed, AER and limb bud mesoderm cells were largely separated (Supplementary Figs. 18–22) and mesodermal cells did not show any enrichment scores for AER-related gene sets (Supplementary Fig. 17). Similarly, we failed to detect the AER program in mesodermal cells during frog tadpole limb regeneration (Supplementary Figs. 17–22). In sum, our analyses revealed an axolotl-specific feature where a part of the AER transcriptional program is also present in the mesodermal lineage cells, which reveals distinctive cellular transcriptional programs for limb development, regeneration, and morphogenesis in general.

## Discussion

The AER is central to successful limb development. However, its presence in salamanders that are widely used to study limb regeneration remained controversial. Our work clarifies ambiguous propositions related to this topic, revealing axolotls have cells with AER transcriptional programs with comprehensive cross-species comparisons that moved beyond single-gene investigations. Thus, our results pave the way for a better understanding of the evolution of limb morphogenesis by highlighting the presence of transcriptionally highly similar populations, albeit showing distinct gene expression. Notably, the axolotl AER cells do not appear to express AER-FGFs (*Fgf4*, *Fgf8*, *Fgf9*, *Fgf17*), aligning with previous reports[4,7]. Instead, they express different FGFs (e.g., *Fgf7* and *Fgf18*), stressing the utility of high-throughput and unbiased methodologies for cross-species comparisons. To gain further insights, future studies

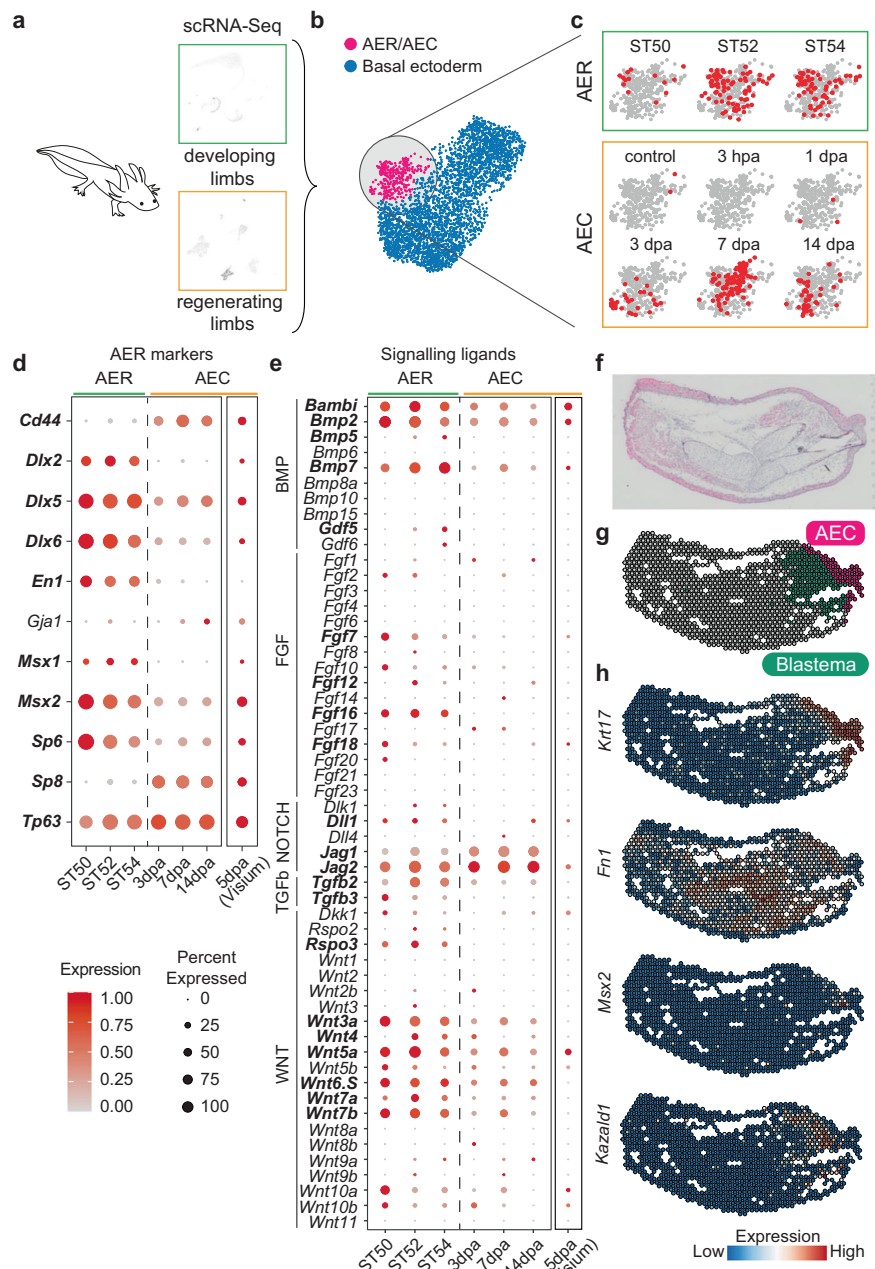

**Fig. 3 | Axolotl AER cells are not entirely re-formed during limb regeneration.**
**a** Schematics describing examples of the used scRNA-Seq datasets of axolotl limb development and regeneration are illustrated. **b** UMAP plot of the basal ectoderm cells of the integrated axolotl limb development and regeneration datasets. Subclustered cell identities are labeled by different colors and text. **c** Sample contribution to the integrated AER/AEC cluster from **b**. Red dots indicate cells from the selected sample; gray dots indicate the other cells in the AER/AEC cluster. hpa: hours-post amputation; dpa: days-post amputation. **d** Dot plot showing AER marker expressions in (left) the scRNA-seq datasets of axolotl limb development and regeneration AER or AEC clusters, respectively, and (right) spatial transcriptomics (Visium) AEC cluster (**g**). The dot color indicates the mean expression that was normalized to the max of each gene; the dot size represents the percentage of cells with non-zero expressions. Please note that the Visium and scRNA-seq datasets were normalized separately within the dataset. Significant differentially expressed genes between AER and AEC were labeled in bold

(two-sided Wilcoxon rank-sum test; *P*-values < 0.05). **e** Dot plot showing signaling ligand expressions in (left) scRNA-seq datasets of axolotl limb development and regeneration AER or AEC cells, respectively, and (right) Visium AEC cluster (**g**). The dot color indicates the mean expression that was normalized to the max of each gene; the dot size represents the percentage of cells with non-zero expressions. Please note that the Visium and scRNA-seq datasets were normalized separately within the dataset. Significant differentially expressed genes between AER and AEC were labeled in bold (two-sided Wilcoxon rank-sum test; *P*-values < 0.05). **f** Hematoxylin and eosin stained 5 dpa axolotl limb regeneration tissue section. The tissue is oriented with the anterior to the top, the posterior to the bottom, proximal to the left, and distal to the right. **g** Clustering of the Visium spots identified known tissue types, including the AEC (pink) and the blastema (green). For the full clustering results, see Supplementary Fig. 15. **h** Expression profiles of selected markers in the AEC and the blastema clusters.

employing genetic cell ablation methods, which are currently unavailable in axolotls, could elucidate the functional role of the identified axolotl AER-like cells during limb development and whether they interact with the underlying mesenchyme as in other species.

Regardless, our comprehensive evaluation provides compelling evidence that axolotl limb buds harbor cells with AER-like features, demonstrating both significant similarities in transcriptional programs and a substantial, albeit not identical, spatial organization.

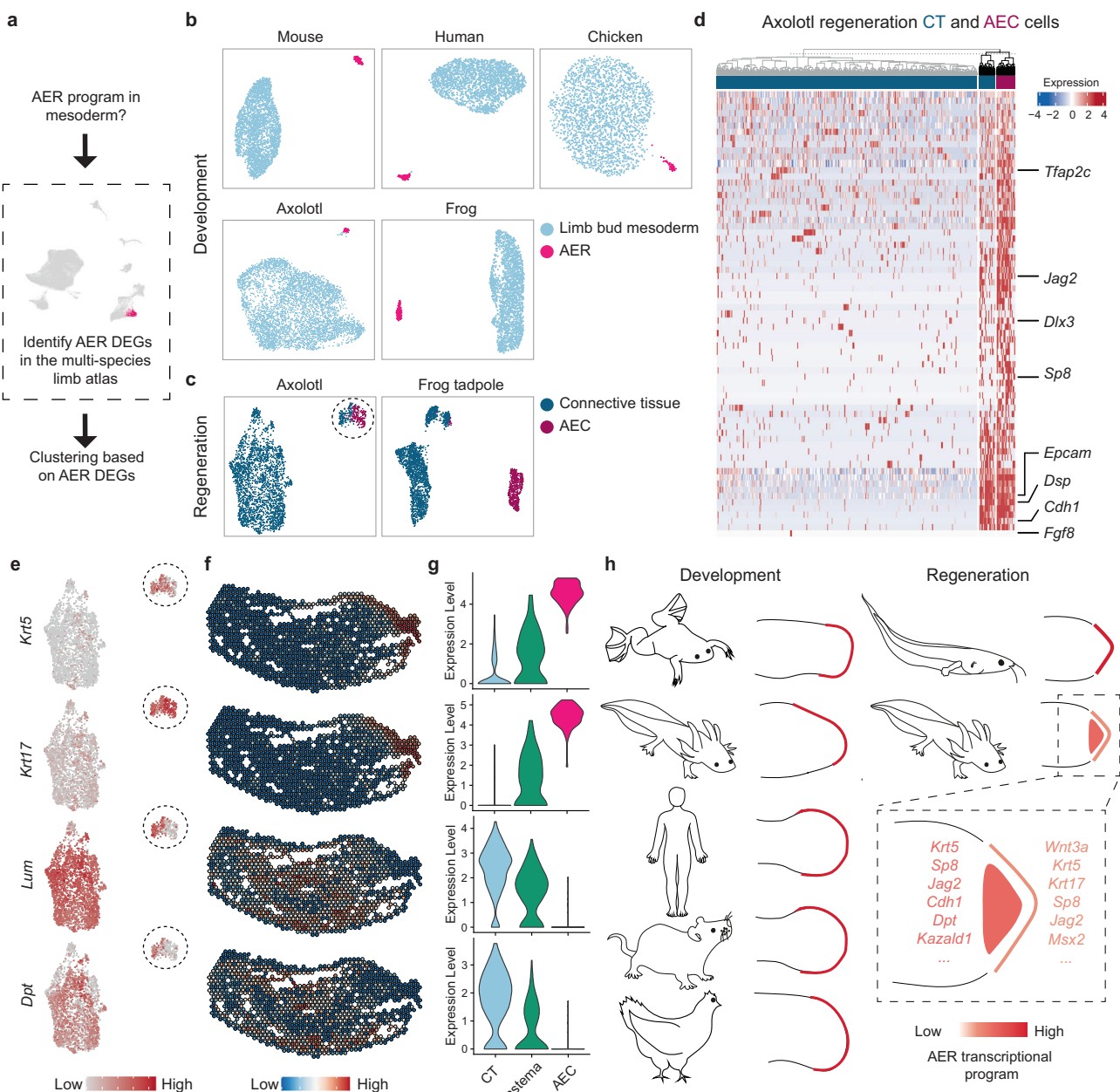

**Fig. 4 | Axolotl limbs contain mesodermal cells showing part of the epithelial AER transcriptional program. a** Schematics of the strategy to evaluate the mesodermal cells showing the AER transcriptional program. AEC and connective tissue (CT) cells are clustered based on differentially expressed genes (DEGs) in the AER cluster of the multi-species limb atlas. **b** UMAP plots of the clustering based on the DEGs in the AER cluster of the multi-species limb atlas (Fig. 1b) for development datasets. Mouse E10.5, chicken E4.5, human CS13, frog NF51, and axolotl ST52 UMAPs are shown. Light blue, limb bud mesoderm cells; pink, AER cells. **c** UMAP plots of the clustering based on the DEGs in the AER cluster of the multi-species limb atlas for the regeneration datasets. Dark blue, CT cells; dark pink, AEC cells. CT cells gathered with the AEC and the AEC are circled with a dashed line. **d** Heatmap showing shared gene expressions in the axolotl CT and AEC populations. Dendrogram based on gene expression profile indicates a subset of CT cells shows similarity to the AEC population and is highlighted with black

dendrograms. Note that these shared genes were identified by *K*-means clustering (Supplementary Fig. 22a) of the whole AER gene list used in Fig. 4b, c, except for the manually added *Fgf8* due to its high relevance to AER. **e** The expression profile of the example epithelial and fibroblast genes in the UMAP plot in Fig. 4c axolotl regeneration dataset. **f** Expression profiles of the example epithelial and fibroblast genes are visualized on the Visium dataset. Please note that the *Krt17* data is the same as in Fig. 3h. **g** Violin plot showing expression levels of the example epithelial and fibroblast genes in the CT, the blastema, and the AEC clusters of the Visium dataset. **h** Schematics illustrating the AER transcriptional program in developing (axolotl, frog, human, mouse, chicken) (left) and regenerating (axolotl, frog tadpole) limbs (right). Red to orange colors are used to indicate changes in the AER transcriptional program, relevant explicitly in the axolotl regeneration schematics. Dash rectangle indicates the zoom-in view of the axolotl regenerating limb with selected AER genes.

Limb regeneration has long been thought to mimic some aspects of limb development[3,17,37,38], offering a roadmap to establish strategies to regrow lost mammalian limbs. Nevertheless, contrary to the commonly accepted assumptions, we found the axolotl AER is not entirely re-formed during limb regeneration. Given the multiple functional

assays demonstrating the essential role of the salamander AEC[12,39–45], our results imply that the distinctive axolotl AEC signaling profile might be sufficient for regeneration. Alternatively, genes, other than the ones commonly associated with well-studied signaling pathways, might be critical for the function of the AEC.

Our work here suggests that more complex and species-specific processes exist for both limb development and regeneration, even among regeneration-competent animals. Notably, axolotl limbs contain parts of the AER program both in ectoderm and mesoderm when compared to other analyzed species - including humans - where only ectodermal cells show an AER program (Fig. 4h), highlighting significant differences for limb morphogenesis across species. Moreover, identifying the epithelial AER program in the axolotl mesoderm demonstrates an uncommon cross-lineage cellular activity, given the previous studies were not able to identify a lineage switch between ectodermal and mesodermal populations[7,46–48], and pose new cell-type evolution and mesodermal plasticity scenarios. Finally, our study provides a non-developmental route to impart limb regeneration to mammals. Inducing epithelial AER programs in CT and ectodermal-derived populations at the same time might be an alternative strategy to regrow lost mammalian limbs.

## Methods

### Animal husbandry
Axolotls (*Ambystoma mexicanum*) husbandry and experimental procedures were performed according to the Animal Ethics Committee of the State of Saxony, Germany. All mice and chicken embryo samples were collected in accordance with the Swiss Federal Veterinary Office guidelines and as authorized by the Cantonal Veterinary Office (cantonal animal license no: VD3652c, and national animal license no: 33237). No sex determination was performed in any of the used animal samples in this study.

Axolotl husbandry was performed in the CRTD axolotl facility using methodology adapted from Khattak et al.[49] and according to the European Directive 2010/63/EU, Annex III, Table 9.1. Axolotls were kept in 18–19 °C water in a 12 h light/12 h dark cycle and a room temperature of 20–22 °C. Animals were housed in individual tanks categorized by a water surface (WS) area and a minimum water height (MWH). Axolotls of a size up to 5 cm SV were maintained in tanks with a WS of 180 cm$^2$ and MWH of 4.5 cm. Axolotls up to 9 cm SV were maintained in tanks with a WS of 448 cm$^2$ and MWH of 8 cm.

Pregnant CD-1 mice were obtained from Charles River Laboratories. The mice were housed in rooms with a regular dark/light cycle and fed a standard rodent diet and water ad libitum. Chickens were outbred and chicken eggs were obtained from a local farm (Brüterei Stöckli AG). Eggs were incubated at 38 °C and staged according to the Hamburger–Hamilton staging chart.

### scRNA-Seq data acquisition and preprocessing
Publicly available raw sequencing data, and the expression matrices of axolotl-developing limb datasets were downloaded (Supplementary Data 1). The developmental stages of each sample were determined by the original studies including humans (Carnegie Stage 13), mice (embryonic day (E) 9.5, E10.5, E11.5, E12.5, E16.5), chickens (Hamburger–Hamilton stage 25), frog tadpoles (Nieuwkoop and Faber (NF) stage 50, NF51, NF52 NF54), and axolotls (stage 50, 52, 54).

For humans, mice, chickens, and frogs datasets, CellRanger (v6.1.1.1) was used for preprocessing the raw data. CellRanger *mkref* function was used to build references for each species with corresponding genome sequences and annotation files (Supplementary Data 1). CellRanger count was used to identify valid cell barcodes, align reads, and quantify gene expression. For axolotl regeneration datasets, kallisto|bustools[50] was used to generate the expression matrix. Cells passed the filtering in both *DropletUtils::emptyDrops* and *DropletUtils::defaultDrops* were retained for further analyses[51,52]. The default setting was used unless noted. Cells were further filtered based on dataset-specific thresholds of three metrics: the mitochondrial percentage, the number of transcripts, and the number of genes (Supplementary Data 1).

### Gene list curation of the five analyzed species for comparison
Human, mouse, and chicken orthologs were downloaded from BioMart. As the frog (*Xenopus laevis*) is a pseudo-tetraploid animal, a pseudo-genome was generated: alleles between L and S homologous that are showing the higher expression were considered as the expression[23]. Axolotl genes with the same names as human genes were defined as the orthologs. Multiple axolotl transcripts, thereby genes, could be annotated with the same human gene, for which, only the one with the maximum expression was considered as the expression. In total, 8855 genes were retained for Fig. 1b–d, and Supplementary Figs. 3a, b and 13.

### Curation of the gene sets
For cell cycle-related genes, the human gene list was retrieved from a previous study[53] and orthologs were used for the other species. For signaling ligands, human and mouse genes encoding ligands of FGFs, WNTs, TGFbs, NOTCH/DELTAs, and BMPs signaling pathways were retrieved from CellChat[54], and orthologs were used for chicken, frog, and axolotl (Supplementary Data 2).

### Clustering of individual scRNA-Seq datasets
Seurat (v4.0.3)[55] was used for clustering the scRNA-Seq datasets individually. Briefly, the expression matrix was normalized and scaled. The top 2000 highly variable genes were used for principal component analysis (PCA). The first 15 principal components were chosen to build *K* nearest neighbors (KNN) graph and Louvain clustering. Data were visualized using 2-dimensional UMAP. The default setting was used unless noted. Cell cycle correction was performed when the clustering was biased by the cell cycle (Supplementary Data 1). For this, cell cycle scores were first assigned to each cell using *CellCycleScoring* in Seurat with species-specific cell-cycle genes. Then, the absolute weights of the cell cycle genes (loading values) were summed for each principal component (PC). PCs with values exceeding dataset-specific thresholds were defined as cell cycle-correlated PCs (Supplementary Data 1). PCA was re-run without the top 10% genes from cell cycle-correlated PCs. Specifically, one cluster of low read count in the chicken HH25 dataset was removed from future analysis.

### Integration of datasets using Seurat
For the multi-species limb atlas, individual datasets were processed into Seurat objects where only one-to-one orthologs of the five species were retained. The annotated cell clusters from each dataset were downsampled to half when exceeding 500 cells in number. The sctransform-based normalization (*SCTransform*) was performed. Genes were ranked by the number of datasets they are deemed variable in (*SelectIntegrationFeatures*) and the top 3000 genes were used to integrate all the datasets (*FindIntegrationAnchors* and *IntegrateData*). Clustering was corrected for the cell cycle effect as described using 1.5 as the threshold to identify cell cycle-correlated PCs. Annotation was performed based on cell-type specific marker gene expression (Supplementary Fig. 3d). Specifically, one confounding cluster spreading across the whole UMAP was removed. The remaining cells were used to determine the final KNN graph (k = 30) and UMAP with the top 30 PCs.

For clustering without the chicken dataset (Supplementary Fig. 3h), the same parameters were used as above. For clustering with mouse E16.5 proximal limb buds (Supplementary Fig. 3g), the same pipeline was used except that k.weight was set to 500 during integration and threshold as 3 in cell cycle removal.

For the axolotl regeneration dataset, individual datasets of different time points were integrated into a Seurat object as described without cell downsampling (Supplementary Fig. 13).

## Integration of datasets using SAMap

SAMap (version 1.0.12) was used[31]. The same cells used for Seurat integration were used for SAMap integration. Briefly, pair-wise tblastx (version 2.9.0) was performed for transcriptomes of five species using the provided map_genes.sh. BLAST bit scores were used as the initial gene-gene weights. Then, datasets from the same species were concatenated as one input of *samap.run* to perform iterative clustering. In each round of clustering, the gene-gene weights were updated as expression correlations of the matched cells until the alignment score was above the default threshold. The default setting was used unless noted. Cell clusters were defined using the Louvain algorithm implemented in SCANPY[56] with 2.0 resolution.

## Calculation of integration accuracy

To calculate the integration accuracy, cell type annotations derived from Seurat and SAMap integrated atlases were compared to the annotation of individual dataset clustering results. For this, confusion matrices were generated, using *confusion_matrix* in the cvms R package (1.3.8). The confusion matrices were normalized to the maximum cell number of each cell cluster in the integrated atlases to normalize sample sizes.

## Cluster to lineage and cell type annotation

The lineage or cellular identities of Louvain-defined clusters were defined based on the expression of literature-supported markers (Supplementary Figs. 3d, 12c, 13d, 15e). Ambiguous clusters were labeled as "Unknown". Specifically in the axolotl regeneration dataset (Supplementary Fig. 13b), the AEC population was determined based on the annotation from the basal ectoderm subclustering (Fig. 3b, c) where cells from regeneration datasets that fell into the AEC/AEC cluster were defined as AEC. Additionally, the differentially expressed genes (DEGs) of each cell cluster were identified using *FindMarkers* with the default setting and were visualized in Supplementary Figs. 3d, 12c, 13d, and 15e.

## Transcriptome-wide comparison of cluster similarity across species

MetaNeighbor was used to calculate the cluster similarity[57]. MetaNeighbor scores cluster similarity with AUROC (area under the receiver operating characteristic). The range of the AUROC scores is from zero to one, where zero indicates dissimilar and one indicates similar. Of note, a value of 0.5 means the algorithm is not able to decide the similarity. The 303 variable genes identified by *MetaNeighbor::variableGenes*, the top 3000 variable genes identified by *Seurat::FindVariableFeatures*, and the top 50 PCs from *Seurat::RunPCA* in the multi-species limb atlas (Fig. 1b) were used as input in unsupervised mode, respectively (Fig. 1f and Supplementary Fig. 6).

## Single-cell gene set enrichment analysis (scGSEA) of the signaling ligands

AUCell[58] was used for scGSEA. Signaling ligands for each species and the raw count expression of each developing limb dataset were used as input (Supplementary Data 2). AUCell scores were averaged for each species and developmental stage for visualization in Fig. 1g and Supplementary Fig. 7a. The average scores were further normalized to the maximum of each dataset and shown in Supplementary Fig. 7b.

## Identification of axolotl AER marker gene expression

In the axolotl developing limbs datasets (Supplementary Fig. 2e), the axolotl AER cells were compared to all the other cells and to only the ectodermal cells using *FindMarkers* with the default setting. Axolotl AER cells upregulated genes in all comparisons were intersected, resulting in 14 shared genes (Supplementary Fig. 9a). By visually checking the expression in each cluster, *Dr999-Pmt21178* and *Vwa2*

were determined as the most specific ones out of the shared genes (Supplementary Fig. 9b).

## Hybridization chain reaction (HCR) on whole-limb samples

All animal samples (axolotl stages 46, 50, and 53, mouse E10.5, and chicken HH22) were fixed with 4% formaldehyde in 1×PBS for 40−60 min and stored in 100% ethanol at −20 °C. Fixation was carried out on a rotator at room temperature. Limbs were dissected and HCR protocol was applied as described previously[59] with modifications. Briefly, limbs were transferred in a new Eppendorf tube containing 500 µl of wash buffer (Molecular Instruments) that has been incubated for 10 min at 37 °C. The supernatant was removed and replaced by 500 µl pre-heated hybridization buffer (Molecular probes) for a 30 min incubation at 37 °C. During this incubation, the probe solution was prepared by diluting mRNAs targeting probes to 40 nM in 300 µl hybridization buffer and incubated for 30 min at 37 °C. Probes for the axolotl *Vwa2, Dr999-Pmt21178, Msx2, Epcam, Fgf7,* and *Fgf18,* and mouse and chicken *Fgf8* and *Vwa2* were designed based on the transcript sequences obtained from the matched transcriptome used in the data analysis (Supplementary Data 1), were purchased from Molecular instruments. The hybridization buffer from samples was taken out and probe solution was placed on samples for a 12 h incubation at 37 °C. The samples were then washed twice for 30 min with wash buffer and twice for 20 min with 5×SSC-T on a rotator at room temperature. To visualize probes, amplification solution was prepared by first heating the fluorophore attached hairpins pairs (Molecular instruments) that match the probes to 95 °C for 90 s. Hairpins were then left in the dark at room temperature for 30 min. Afterwards, the final amplification solution was prepared at 72 nM h1 and h2 in 250 µl amplification buffer. Samples were first incubated in amplification buffer without hairpins for 10 min, then placed in the final amplification solution at room temperature, protected from light, for 12−16 h on a rotator. Samples were washed with 2 × 20 min SSC-T and incubated in 20 µM Hoechst (Sigma, 2261) diluted in 1×PBS at room temperature in the dark for 30 min. Finally, the samples were washed 3 × 10min with PBS and mounted as described below. All staining experiments were performed with at least 3 biological replicates except chicken and mouse staining experiments, which were performed with 2 biological replicates. Each biological replicate contained at least 2 technical replicates. (see statistics and reproducibility for full information). Representative images are shown in the manuscript.

## Whole-mount HCR samples imaging

Whole limbs were mounted in 0.8% ultra-low gelling temperature agar (Sigma, A5030) in 1×PBS. Confocal imaging was performed using Leica SP8 inverted confocal microscope with 10x HC PL Fluotar, 20×/0.75 HC PL Apo air or 40×/1.25 HC PL Apo air objective and post-processing was performed using ImageJ software. Fiji was used for maximum projection of *z*-stacks and to adjust contrast to highlight biological relevance. If needed, images were cropped, flipped and/or rotated to highlight biological relevance.

## Subclustering of the basal ectoderm from axolotl developing and regenerating limbs

Basal ectodermal cell clusters, including the AER clusters, from both axolotl developing limbs (Supplementary Fig. 2e) and regenerating limbs (Supplementary Fig. 13b) were extracted and integrated as one Seurat object as described. Cell cycle correction was performed using 1 as the threshold to define cell cycle-correlated PCs. After removing one cluster with connective tissue markers (*Prrx1*), the KNN graph and the UMAP analysis were re-run using the top 30 PCs (Fig. 3b).

## Comparisons between AER and AEC cell clusters

In Fig. 3d, e, the two-sided Wilcoxon rank-sum test was performed on AER ($n = 157$) and AEC ($n = 175$) cells. *P*-values were corrected with the

Benjamini-Hochberg method. Genes with adjusted *P*-values less than 0.05 were considered as significant. Further, DEGs of AER and AEC were calculated using *FindMarkers* function with default parameters (Supplementary Data 3). Functional enrichment analysis was performed on the top 200 DEGs (ordered by fold change) of each cell population via Metascape[60] (Supplementary Fig. 14b).

### Spatial transcriptomics
Spatial transcriptomics was performed using the Visium Spatial Gene Expression System (10× Genomics, Pleasanton, CA). Animals 8–9 cm snout to tail were amputated at the level of the lower arm and allowed to regenerate until 5 dpa. Limbs were then harvested at the level of the upper arm, fresh frozen in OCT, and then stored at −80 °C. Samples were sectioned at −20 °C at a thickness of 10 μm. Optimization and gene expression assays were carried out according to the manufacturer's protocol. Briefly, slides were fixed in −20 °C methanol, dried with isopropanol, and stained with H&E. A tile scan of all capture areas was generated using an Olympus OVK automated slide scanner system with a color camera and fluorescent module.

For tissue optimization, enzymatic permeabilization was conducted for 0–30 min, followed by first-strand cDNA synthesis with fluorescent nucleotides. The slide was reimaged using the standard Cy3 filter cube. An optimal permeabilization time of 20 min was determined by visual inspection to maximize mRNA recovery while at the same time minimizing diffusion. For gene expression, the initial workflow was similar to the optimization procedure. Library preparation, clean-up, and indexing were conducted using standard procedures. Samples were subjected to pair-ended sequencing using an Illumina Multiplexing generating ~75 M reads.

### Spatial transcriptomics data analysis
Gene expression quantification of the Visium dataset was done using kallisto (0.48.0) and bustools (0.41.0) as previously described[61]. Cells were filtered with *DropletUtils::emptyDrops* and *DropletUtils::defaultDrops* as described. SpaceRanger (v2.0.0) was used to match the Visium spots to the tissue slice. Seurat was used for sctransform-based normalization, PCA, and clustering with the top 30 PCs.

### Identification of AER-specific modules
To define AER cell identity-related transcriptional modules, consensus non-negative matrix factorization (cNMF) (v1.4) was used[33]. The normalized expression matrix with the connective tissue and AER cells from the multi-species limb atlas was used as the input. cNMF requires manually defining the number of modules (controlled by parameter *k*). We tested *k* values from 5 to 17, among which *k* = 13 gave the lowest error and decent stability (Supplementary Fig. 16b). Modules 11, 12, and 13 were scored specifically high in only AER cells (Supplementary Fig. 16c) and were defined as AER-specific programs and used for further analyses in Supplementary Figs. 19–21.

### AER programs in mesodermal lineage cells during limb development and regeneration
To examine if the AER programs are present in limb bud mesoderm and CT, we first defined four gene sets representing the AER programs: (1) DEGs in the AER cluster of the multi-species limb atlas (Fig. 1b), which were calculated using *FindMarkers* with the default setting. In total, 545 genes with adjusted *P*-values less than 0.05 and were considered significantly upregulated in multi-species AER cluster. (Supplementary Data 4). (2) three cNMF-derived AER-specific modules that were determined as described above (Supplementary Fig. 16 and Supplementary Data 4).

First, scGESA was performed using AUCell as described using the top 200 genes in the AER DEGs list. AUCell scores were visualized in the UMAP for each dataset (Supplementary Fig. 17). Second, clustering with the top 500 genes from the four AER-related gene sets was

performed on each developing and regenerating limb dataset. Only the AER and the limb bud mesoderm clusters in development samples, or the AEC and the connective tissue clusters in regeneration samples, were extracted. Then, PCA was performed using each gene set, respectively. The top 30 PCs were used to project the data into the UMAP (Supplementary Figs. 17–21).

### Gene ontology analysis of the cNMF-derived AER modules
The top 100 genes from each module were used to perform gene ontology analysis using clusterProfile[62] (Supplementary Fig. 16c). *P*-values were corrected using the Benjamini-Hochberg method. GO terms with adjusted *P*-values less than 0.01 were considered as the significantly enriched GO terms.

### Statistics and reproducibility
Replicates for the HCR experiments are reported as follows. Technical replicates (total number of different animals used) are denoted as *n*, and independent replicates (total number of experiments performed on separate days, and/or with different batches) are denoted as *N*. For stage 46 axolotls, *n* = 20, *N* = 7 (*Dr999*); *n* = 16, *N* = 5 (*Vwa2*); *n* = 6, *N* = 2 (*Msx2*, *Epcam*); *n* = 10, *N* = 2 (*Fgf7*, *Fgf18*). For stage 50 axolotls, *n* = 5, *N* = 2 (*Dr999* and *Vwa2*). For stage 53 axolotls, *n* = 20, *N* = 7 (*Dr999*); *n* = 16, *N* = 5 (*Vwa2*); *n* = 4, *N* = 2 (*Msx2*, *Epcam*). For E10.5 mice, *n* = 15, *N* = 4 (*Fgf8*); *n* = 10, *N* = 3 (*Vwa2*). For HH22 chicken, *n* = 4, *N* = 2 (*FGF8*, *VWA2*).

No statistical method was used to predetermine the sample size. No data were excluded from the analyses. The experiments were not randomized. The investigators were not blinded to allocation during experiments and outcome assessment.

### Reporting summary
Further information on research design is available in the Nature Portfolio Reporting Summary linked to this article.

## Data availability
The axolotl Visium raw data are publicly available on Gene Expression Omnibus (GEO) under the accession code GSE243225. Previously published scRNA-seq data that were re-analyzed here are from NCBI GSE143753 (CS13 human limb buds), NCBI GSE157329 (CS13 human limb buds), NCBI GSE137335 (E9.5 mouse limb buds), NCBI GSE158820 (E10.5, E11.5, E12.5 mouse limb buds), NCBI GSE130439 (HH25 chicken limb buds), NCBI GSE165901 (NF 50, NF51, NF52 frog limb buds; Stage 50, 52, 54 axolotl limb buds), ArrayExpress E-MTAB-9104 (NF 54 frog limb buds, 5 days post-amputation NF 52 frog regenerating limbs), NCBI PRJNA589484 (0–14 days post-amputation axolotl regenerating limbs). Requests for raw image data should be addressed to C.A. Source data are provided in this paper.

## Code availability
Custom scripts are available at https://github.com/AztekinLab/Axolotl_AER_2023.

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

## Acknowledgements

We thank Aztekin and Sandoval-Guzman Labs for the discussions; M. Gurel and M. Brbic for discussions on factorization and module identification; M. Ros, D. Suter, G. La Manno, and M. Gurel for their critical reading of the manuscript; A. Petzold and the CMCB Genome Facility for help with spatial transcriptomics sequencing and data processing. C.A. is supported by EPFL School of Life Sciences ELISIR Scholarship, the Foundation Gabriella Giorgi-Cavaglieri, Branco Weiss Fellowship, and SNSF NRP79 (407940-206349). R.A is supported by a Von Humboldt Foundation Research fellowship PRT 1208176 HFST-P and a DFG Eigene Stelle Grant AI 214/1-1.

## Author contributions

Conceptualization: mainly C.A., J.Z.; Methodology: C.A., J.Z.; Software: J.Z.; Formal analysis: J.Z.; Investigation: C.A., J.Z., G.T., E.S., R.A.; Project administration: C.A.; Supervision: mainly C.A., T.S.G.; Funding acquisition: C.A., T.S.G.; Resources: K.B.; Data curation: C.A., J.Z.; Writing—original draft: mainly C.A., J.Z.; Writing—review and editing: all authors.

## Competing interests

The authors declare no competing interests.
