## [Peer Review File · Nature Communications]

Multi-species atlas resolves an axolotl limb development and regeneration paradoxReviewer #1 (Remarks to the Author):

Zhong and colleagues have investigated whether axolotls contain an apical ectodermal ridge (AER), a key signalling centre, during development and whether a similar population of cells is present during regeneration using transcriptomic data. The authors describe two significant observations; (a) cells with AER-like characteristics are present in the developing axolotl limb; and (b) that mesodermal cells display partial AER-like characteristics during regeneration. Both of these findings represent a clear advancement to our current knowledge and would be of broad interest to developmental biologists and the regenerative medicine communities. The first observation challenges a long-held belief that during amphibian evolution salamanders developed an alternative mechanism of limb development that did not rely on the presence of an apical ectodermal ridge (Sturdee and Connock, 1975; Tank et al., 1977). The second, highlights the importance of the mesoderm in contributing to pattern formation through signalling mechanisms. The majority of complex tissue regeneration studies attribute signalling properties to epidermal derived tissues such as the wound epidermis and this demonstrates an important paradigm shift. Overall, I am supportive of the concepts and work described in this manuscript, however at present, many of the findings are observational and the manuscript could benefit from some additional evidence to support the claims (as detailed below).

1. A strength of the manuscript is that the author's used multiple integration methods to support their argument that axolotl limb buds contain cells with AER-like characteristics. To further support this data, it would be useful if they could integrate data from a developing limb in which the AER is not present (e.g. prior to AER formation or after the AER has regressed) as a negative control. This would ensure that the integration methods they use preserve both cross-species similarities as well as differences and do not over-correct the data.
2. The quality of the data obtained from the developing chick limb bud appears to be lower than the other species (i.e. less transcript numbers per cells, less genes per cell and a higher percentage of mitochondrial genes per cell) as per Extended Figure 1. How did this affect the data integration? Were similar results obtained without this dataset?
3. On line 70, the authors state that the large majority of axolotl cells in this cluster co-express many of the AER markers. Can the authors provide a percentage? (i.e. 98% express at least 5 or more AER markers).
4. In extended Figure 2 E, the developing axolotl limb at stage 50 does not appear to have any annotated AER cells however in Figure 1E, AER markers are found in some ectodermal cells at this stage. Do these cells cluster with another population? How do the authors account for these differences?
5. The authors utilize Dr999-Pmt21178 and Vwa2 to mark the axolotl AER. These two markers don't appear to co-localize as much as expected. It would therefore be useful to show co-expression of the axolotl-like AER cells with other general AER markers also via whole-mount hybridization chain reaction (or IHC) that were identified in 1E (some of them) and how these compare to other species (e.g. mouse or xenopus).
6. In Figure 2G/H, the authors present spatial transcriptomic data using the Visium platform. The resolution of each spot does not equate to a single cell. How did the authors account for this in their interpretation of the data? Were any computational methods employed to resolve this issue? (e.g. Use of Cell2Location or STDeconvolve).
7. The experimental methods or Figure legends lack information on the number of times experiments were performed (e.g. for hybridization chain reaction – How many animals were observed and are these images representative of that?)
8. Finally, most of the observations, particularly relating to the presence of AER-like cells during axolotl limb development are correlative. If the authors could provide any evidence that these cells function as an AER (i.e. experimental evidence) this would really strengthen the paper.

Reviewer: Mekayla Storer

Reviewer #2 (Remarks to the Author):

- What are the noteworthy results?

The present study has integrated and analyzed several previously generated vertebrate limb development/regeneration single cell transcriptomics datasets and have 3 major conclusions:

- 1) Axolotls possess Apical Ectodermal Ridge (AER) like cells during limb development.
- 2) Apical Ectodermal Cap (AEC) formed during axolotl limb regeneration and AER are distinct.
- 3) Axolotl mesoderm expresses AER factors.

- Will the work be of significance to the field and related fields? How does it compare to the established literature? If the work is not original, please provide relevant references.

Limb development studies have mostly been restricted to amniotes like chicken and mice that do not have the capacity to regenerate their limbs. Studies involving naturally occurring mutants, tissue excisions/transplantations and genetic ablations have identified the Apical Ectodermal Ridge or AER as one of the important signaling centers that triggers proximal-distal limb outgrowth. Fgfs like Fgf 4, 8, 9 and 17 are hallmarks of the AER and are key drivers of limb outgrowth. The AER-Fgfs also form a feedback loop with mesenchymal Shh and Grem1 to trigger limb outgrowth.

Salamander species like axolotls and newts are popular vertebrate models capable of limb regeneration. The question whether limb regeneration is a recapitulation of limb development is a long standing one. While there have been sporadic studies on limb development in salamanders, these have only explored a handful of genes and their functions. This study uses previously generated datasets to present an elaborate panel of markers for various tissue types involved during vertebrate limb development and specifically focuses on the ectodermal subset, AER cells to conclude that axolotls possess an AER. While this study is a good comparative analysis of various vertebrate models relevant to limb literature and for the first time uses spatial transcriptomics approach in axolotl limb regeneration paradigm, the first conclusion is overstated and needs substantive validations. The absence of canonical AER-Fgfs in the "axolotl-AER" and lack of any functional testing works against the claim. The authors also need to explore the placement of the feedback loop and interaction between the ectoderm and the mesenchyme during limb outgrowth in axolotls.

The finding that AEC and AER are distinct is novel and partially answers the paradox whether limb regeneration is a recapitulation of limb development from the ectodermal cell perspective alone. The finding that the axolotl mesoderm possesses AER factors has been previously reported in axolotl limb development (Purushothaman et al., 2019; Glotzer et al., 2022) and axolotl limb regeneration (Nacu et al., 2016) albeit based on a handful of markers, and this study further extends the panel of genes with few validations based on spatial transcriptomics.

- Does the work support the conclusions and claims, or is additional evidence needed?

The conclusions and claims here are based on single cell transcriptomics analyses with validation of few genes using HCR FISH and Visium.

For axolotl limb development, HCR FISH validation is only done for 2 genes Dr999 and Vwa2 that do not have comparative expression profiles in mouse and chicken limb literature. HCR FISH for these genes on mouse or chicken can partially support the claim. Additionally, Dr999 expression at stage 50 and 52 (extended data Fig.9 c) is not convincing as it does not show the classic AER like domain and is present throughout the ectoderm.

Previous studies (Purushothaman et al., 2019, Ghosh et al., 2008) show mesenchymal expression for genes like Msx2 and Wnt5a, while single cell transcriptomics data here shows ectodermal expression. The authors must depict axolotl specific, all cell type UMAP plots and violin plots (similar to extended data Fig.9 b) for all the genes in the AER specific list and provide HCR FISH/Visium validations for highly expressed genes to confirm exclusive expression in the AER.

- Are there any flaws in the data analysis, interpretation and conclusions? Do these prohibit publication or require revision?

The methodology used for single cell transcriptomics analysis, HCR and Visium is sound however

the interpretations and conclusions regarding axolotl AER is overstated for reasons mentioned above. The paper requires major revisions to support the claim that axolotls possess an AER, including validation of the expression of AER markers and functional analysis of axolotl ectoderm and its association with the underlying mesenchyme.

- Is the methodology sound? Does the work meet the expected standards in your field?

The methodology for single cell transcriptomics analysis, HCR and Visium is sound and meets the expected standards in the field. One suggestion for imaging whole mount limbs is to also have the lateral view images to verify if the *Dr999* and *Vwa2* are expressed at the dorsal-ventral border of the limb buds/digits.

- Is there enough detail provided in the methods for the work to be reproduced?

The methodology is elaborate enough and can be reproduced.

We thank the reviewers for their support of our work and insightful comments, which substantially improved the manuscript and at the same time made us realize that we should have provided more details about the strengths and limitations of our findings and used methodologies. In light of these, we have addressed all comments through additional experiments, re-wording of the current manuscript, and by providing additional clarifications. In this document, we show reviewer comments in bold blue, our response in black, the figure legend in grey and the revised manuscript text in green italics below.

During the revision process, we identified a minor error in certain figures pertaining to the annotation of goblet cells mixed with differentiating ectoderm. It is important to note that this mistake does not affect the results of the analyzed data or our interpretations. We have rectified the error and updated the related figures (Fig. 1b, Supplementary Figs. 3a and 3c, Supplementary Fig. 7). We apologize for this mistake. Secondly, one of the authors (Rita Aires) updated her affiliation and associated funding agency. We have now updated this information in the document.

REVIEWER COMMENTS

Reviewer #1 (Remarks to the Author):

Zhong and colleagues have investigated whether axolotls contain an apical ectodermal ridge (AER), a key signalling centre, during development and whether a similar population of cells is present during regeneration using transcriptomic data. The authors describe two significant observations; (a) cells with AER-like characteristics are present in the developing axolotl limb; and (b) that mesodermal cells display partial AER-like characteristics during regeneration. Both of these findings represent a clear advancement to our current knowledge and would be of broad interest to developmental biologists and the regenerative medicine communities. The first observation challenges a long-held belief that during amphibian evolution salamanders developed an alternative mechanism of limb development that did not rely on the presence of an apical ectodermal ridge (Sturdee and Connock, 1975; Tank et al., 1977). The second, highlights the importance of the mesoderm in contributing to pattern formation through signalling mechanisms. The majority of complex tissue regeneration studies attribute signalling properties to epidermal derived tissues such as the wound epidermis and this demonstrates an important paradigm shift. Overall, I am supportive of the concepts and work described in this manuscript, however at present, many of the findings are observational and the manuscript could benefit from some additional evidence to support the claims (as detailed below).

We express our gratitude to the reviewer for recognizing and acknowledging the value of our research. Indeed, unlike previous investigations that were confined to morphological observations or limited gene analyses, our work employed a comprehensive and quantitative cross-species comparison. This approach allowed us to re-evaluate whether the axolotl limb buds contain a population similar to the AER. Surprisingly, our findings, as pointed out by the reviewer, revealed that the signaling profile associated with regeneration is not predominantly observed in the regenerating axolotl epidermis but, instead, is mostly found in the connective tissue lineage, providing a fresh perspective for future studies in regeneration.

1. A strength of the manuscript is that the author's used multiple integration methods to support their argument that axolotl limb buds contain cells with AER-like characteristics. To further support this data, it would be useful if they could integrate data from a developing limb in which the AER is not present (e.g. prior to AER formation or after the AER has regressed) as a negative control. This would ensure that the integration methods they use preserve both cross-species similarities as well as differences and do not over-correct the data.

We thank the reviewer for pointing out this oversight which gives us an opportunity to address this critical point.

In our original submission, we have already included results that demonstrate the discriminative power of our methodology in identifying both species-conserved and species-specific cell types. Specifically, amphibian skin at analyzed stages is known to contain significant amounts of goblet cells (i.e. PMID: 30692997, 35879314). Importantly, in the analyzed datasets, these cells are predominantly found in amphibians such as frogs and axolotls, with minimal representation (~2.3% in total) in mammals and chickens (Supplementary Fig. 3a, 3b,

and below). This finding underscores the preservation of species differences within our integrated dataset, thereby supporting the notion that the identification of axolotl cells within the AER cluster is not an artefact resulting from species over-correction. We now added this result in Supplementary Fig 3a and revised the text accordingly.

Pie chart showing the percentage of cells from each species and developmental stages in the Goblet cells cluster in the Seurat-integrated UMAP in Fig. 1a.

Revised text line 60. "Critically, the goblet cell cluster, that is prevalent in amphibian skin, was dominated by frog and axolotl cells, with minimal representation (~2.3% in total) of mouse, human, and chicken cells (Supplementary Fig. 3f), emphasizing that the established multi-species atlas preserves species-specific variances."

Additionally, in order to address the reviewer's concern more comprehensively, we have incorporated their suggested analysis into our study. Specifically, we have included a publicly-available mouse E16.5 proximal limbs (stylopods) (Zhang et al., bioRxiv, 2023, data accession: E-MTAB-10514) in our integrated dataset, following the same pipeline outlined in our original manuscript (Method, "Seurat" section). Our analysis revealed that 0 out of 1772 cells from the newly added negative control was assigned to the AER cluster (see below).

Barplot showing the ratio of identified AER cluster to the whole basal ectoderm cluster in indicated species and developmental stages.

To emphasize the strength of our methodology, we have revised the main text accordingly.

Revised text line 63: “Furthermore, when we integrated an E16.5 stylopod dataset, which should contain no AER cells, as a negative control, we detected no contribution to the AER cluster from this dataset (Supplementary Fig. 3g), suggesting that our approach provided robust integration with no detected over-correction issue.”

2. The quality of the data obtained from the developing chick limb bud appears to be lower than the other species (i.e. less transcript numbers per cells, less genes per cell and a higher percentage of mitochondrial genes per cell) as per Extended Figure 1. How did this affect the data integration? Were similar results obtained without this dataset?

The chicken dataset in our study does indeed exhibit lower quality compared to other datasets. To address this issue, throughout this study, we implemented the SCTransform normalization method (PMID: 31870423), which effectively accounts for technical factors, including sequencing depth, as a covariate in a generalized linear model. It has been demonstrated to successfully remove the influence of technical characteristics from downstream analyses while preserving biological heterogeneity (PMID: 31870423).

We would like to highlight that one common impact of low-quality cells is their tendency to cluster together. However, this problem is not observed in our dataset (see below). Indeed, cells with different numbers of transcripts/genes and mitochondrial percentages are uniformly distributed across the map, without any bias towards sequencing depths (see below), except for blood cells, which has an expected lower mitochondrial percentage (PMID: 21698761). Additionally, the chicken cells form clusters with cells of the same cell type from other species, as depicted in Supplementary Fig. 3 and 4. These findings demonstrate that our integration pipeline addresses the low-quality issue associated with the chicken dataset. We now added this result as Supplementary Fig 3c.

Violin plots of the number of detected genes (top) and the percentage of expressed mitochondrial genes (bottom) in each cell type in the Seurat-integrated multi-species limb atlas. Cell types were defined in Fig. 1b.

As suggested by the reviewer we now performed clustering without incorporating the chicken dataset, (Methods, "Seurat" section) (see below). We present a zoomed-in view of the AER cluster, highlighting the contribution of axolotl cells (see below, inserted box). Over 95% (116/122) of the axolotl AER-like cells identified in Fig. 1c are identical in the right panel below. These results provide additional evidence that the identification of axolotl AER-like cells remains consistent across different integration approaches. We now added these results as Supplementary Fig. 3f and revised the manuscript text accordingly.

UMAP plot of Seurat-integrated multi-species limb atlas without the chicken dataset. Cells are colored by their lineages and cell type identities. Inserted box: the species contribution to the AER cluster is visualized. Cells from different species are color-coded.

Revised manuscript line 74:

“Moreover, the presence of axolotl cells in the multi-species AER cluster was not affected by the variation in sequencing quality among datasets, as the omission of the chicken dataset (which is of lower quality than other datasets) from the atlas did not change the results (Supplementary Fig. 3h).”

3. On line 70, the authors state that the large majority of axolotl cells in this cluster co-express many of the AER markers. Can the authors provide a percentage? (i.e. 98% express at least 5 or more AER markers).

We thank the reviewer for this comment. It is indeed clearer to give a percentage. We edited the main text accordingly.

Revised manuscript line 80:

“The large majority of axolotl cells in this cluster (~97% expressing at least 3 of the 18 listed AER markers and 85% expressing at least 5) co-expressed many of the AER markers¹, such as Wnt5a and Msx2, whilst some others (e.g., Fgf8) were absent (Fig. 1e and Supplementary Fig. 5), in alignment with recent reports^{4,7”}

4. In extended Figure 2 E, the developing axolotl limb at stage 50 does not appear to have any annotated AER cells however in Figure 1E, AER markers are found in some ectodermal cells at this stage. Do these cells cluster with another population? How do the authors account for these differences?

We apologize for the oversight.

The number of AER cells is relatively low in the stage 50 sample, making them visually difficult to identify, and they are co-localized with ectodermal cells. To rectify this issue caused by the low resolution of the images, we have included a zoomed image of the ectoderm cluster (see below), where the AER-like cells are now highlighted in magenta. These cells correspond to the same population shown in Fig. 1e.

Furthermore, it is important to note that the stage 50 cells do not exhibit a visually distinct localization on the UMAP plot, unlike those in stages 52 and 54. We attribute this to the limited number of cells in the stage 50 sample, as illustrated in Supplementary Fig. 2E. The clustering algorithm requires a sufficient number of cells to accurately model gene expressions and determine distinct localizations. Indeed, we have addressed this issue by considering the localization of stage 50 AER cells in the context of our multi-species atlas, which contains a larger number of AER cells. As shown in Supplementary Fig. 3a, 3b, and 4c, the AER-like cells from stage 50 axolotls were appropriately identified within the AER cluster, alongside AER cells not only from axolotls but also from other species.

We have replaced the original figure in Supplementary Fig. 2e (stage 50) with the below figure to ensure the accurate representation of stage 50 AER-like cells in our analysis.

UMAP visualization of individual limb development dataset for stage 50 axolotl. Left: this plot is the same figure as that in the original version with a zoomed-in view of the ectoderm population in the upper left corner; Right: a further zoomed-in image of the ectoderm population.

5. The authors utilize *Dr999-Pmt21178* and *Vwa2* to mark the axolotl AER. These two markers don't appear to co-localize as much as expected. It would therefore be useful to show co-expression of the axolotl-like AER cells with other general AER markers also via whole-mount hybridization chain reaction (or IHC) that were identified in 1E (some of them) and how these compare to other species (e.g. mouse or xenopus).

Thanks to both reviewers' comments, we realized that our explanation about these marker genes was not sufficient. Firstly, we see neither of these genes to be exclusively expressed in axolotl AER cells nor we reported that these two genes are co-expressed in the same single cell with 100% overlap. scRNA-Seq data showed that low *Vwa2* and *Dr999-Pmt21178* expressions are also seen in basal ectoderm, meanwhile high *Vwa2* and *Dr999-Pmt21178* have more specificity for the axolotl AER (Supplementary Fig. 9b and below). Because of this, we have used HCR which can distinguish low and high expression of these genes. To make this point clear, we now plotted the co-expression plot for these two genes for AER (green) and basal ectoderm cells (grey), which directly demonstrates this quantitative aspect, at the same time reveal the presence of single cells that do not co-express these two markers at the same time.

Co-expression of AER-like genes in the ectoderm in axolotl limb buds. Scatter plot (lower left) shows the co-expression pattern of *Dr999-Pmt21178* and *Vwa2*. Each dot represents a cell whose location is defined by its expression of *Dr999-Pmt21178* (x-axis) and *Vwa2* (y-axis). AER-like cells are colored in shades of green while non-AER basal ectoderm cells are grey.

non-AER ectoderm in grey. Density plots representation of the distribution of the *Dr999-Pmt21178* (top) and *Vwa2* (lower right) expression in AER (in green) and non-AER ectoderm (in grey).

We now included the above figure as Supplementary Fig. 9d and edited the text accordingly to emphasize these very important details.

Revised manuscript line 106:

*“Using the scRNA-seq dataset, we identified *Dr999-Pmt21178* and *Vwa2* as the marker genes with high expression specifically in the axolotl AER cells, although these two genes also had weak expression in the non-AER basal ectoderm (Supplementary Fig. 9a, b, d). We then performed whole-mount hybridization chain reaction (HCR), which is a semi-quantitative mRNA visualization method, of these two marker genes on developing axolotl limbs from different stages. *Dr999-Pmt21178* and *Vwa2* showed specific expression at the dorsal-ventral boundary at the limb bud stages, albeit more scattered compared to AER localization in other species, (Fig. 1h and Supplementary Fig 9c), and digit tips during digit forming stages (Fig. 1i and Supplementary Fig. 9c), resembling, but not identical to, the spatial organization of AER in other species^{1,2}.”*

To further address the reviewer's concern, we followed their suggestion and performed additional staining of a known AER marker *Msx2* in axolotls (Fig. 1e). Our staining showed that *Msx2* is co-expressed with *Dr999-Pmt21178* in the distal axolotl epidermis.

Revised manuscript line 114:

*“We then showed that *Dr999-Pmt21178* positive cells also co-express a known AER marker *Msx2* (Supplementary Fig. 10a-b).”*

Additionally, we also sought to examine the expression of *Dr999-Pmt21178* and *Vwa2* in other analyzed species. To do this, we first reviewed our scRNA-Seq dataset and found that *Vwa2* (unlike *Dr999-Pmt21178*, which has no known orthologs with other species analyzed) is expressed at different levels in mouse, frog, and human AER cells (see below). Similar to the axolotl, the expression of *Vwa2* in the AER is higher than in the basal epidermis in these species. We did not detect *Vwa2* expression in any cells in chicken which could be due to the low sequencing quality of this dataset, as the reviewer also noted above. Then, we performed HCR for *Fgf8* (to label AER) and *Vwa2* on mouse and chicken limbs. In alignment with sequencing results, mouse *Vwa2* is expressed in the distal site of the limb bud, labelling both AER and the surrounding basal ectoderm. Similarly, our staining results for chicken *Vwa2* also showed its expression in the AER, although it could also be seen in other ectodermal cells. Altogether, these staining results further confirm the validity of our cross-species comparison and the conclusion about the axolotl limb buds containing cells with the AER transcriptional program.

Vwa2 expression in humans, mice, chickens, and frogs.

- Violin plots showing *Vwa2* expression across all cell types in indicated species and developmental stages based on the analyzed scRNA-Seq datasets.
- Max-projection confocal image of mouse E10.5 hindlimb limb buds stained for *Fgf8*, and *Vwa2* mRNA via HCR. Gray, Hoechst; Magenta, *Fgf8* mRNA; Cyan, *Vwa2* mRNA. Scale bars in all images: 100 μ m
- Max-projection confocal image of chicken HH22 hindlimb buds stained for *Fgf8*, and *Vwa2* mRNA via HCR. Gray, Hoechst; Magenta, *Fgf8* mRNA; Cyan, *Vwa2* mRNA. Scale bars: (Top) 250 μ m and (Bottom) 25 μ m.

We now included these results as new Supplementary Fig. 11 and add a new paragraph.

Revised manuscript in line 119:

*“Since our results revealed *Vwa2* and *Dr999-Pmt21178* as markers for the AER-like cells in axolotls, we then examined their expression in the AER in other species. Unlike *Dr999-Pmt21178* which does not have orthologs in other analyzed species, *Vwa2* is expressed at different levels in the AER of humans, mice, and frogs in scRNA-seq data (Supplementary Fig 11a). Meanwhile, we did not detect any *Vwa2*-expressing cells in the chicken dataset, which may be due to its low sequencing depth (Supplementary Figs. 1 and 11a). Indeed, performing HCR on developing chicken or mouse limb buds confirmed *Vwa2* expression in mouse and chicken AERs (as assayed by *Fgf8* expression) (Supplementary Figs. 11b and c). Thus, these results indicate that employing cross-species comparisons can unveil novel cell type markers.”*

6. In Figure 2G/H, the authors present spatial transcriptomic data using the Visium platform. The resolution of each spot does not equate to a single cell. How did the authors account for this in their interpretation of the data? Were any computational methods employed to resolve this issue? (e.g. Use of Cell2Location or STDeconvolve).

We thank the reviewer for bringing up this very important technical detail. Indeed, the resolution of Visium spot is not at the single cell level. However, our interpretations and conclusions are based on multiple methods, including scRNA-Seq which is at the single-cell level, effectively overcoming this problem.

In our original submission, we presented Visium and scRNA-Seq results in separate columns (Figure 2) to highlight we reached the same conclusion regarding AEC gene expression signature with two separate methods. Likewise, in the section where we explored AER signature in AEC and blastema, we not only used

scRNA-Seq and Visium, but as the third additional evidence, referred to several publications by three different groups (PMID: 9415422, PMID: 23658691, PMID: 23213371) that support our conclusions. Thus, we did not use any of the deconvolution methods, as our conclusions are reached by multiple complementary methodologies.

In addition to our main response, we would like to note that the Visium cell resolution is much less of a problem in axolotls compared to that of human and mouse samples. Axolotl cells are between 20 to 35µm in diameter. The cell size differences can also be compared from the published images of axolotl limb regeneration (PMID: 35587651) and mouse digit tip regeneration (PMID: 31902657, PMID: 36417876).

7. The experimental methods or Figure legends lack information on the number of times experiments were performed (e.g. for hybridization chain reaction – How many animals were observed and are these images representative of that?)

We apologize for this missing information. We now added the replicate information in the methods as follows:

Revised manuscript in line 520:

“All staining experiments were performed with at least 3 biological replicates (experiments performed on different days and with different batches of animals) except chicken and mouse staining experiments, which were performed with 2 biological replicates. Each biological replicate contained at least 2 technical replicates (animal and limb sample used in a single experiment). Representative images are shown in the manuscript. “

8. Finally, most of the observations, particularly relating to the presence of AER-like cells during axolotl limb development are correlative. If the authors could provide any evidence that these cells function as an AER (i.e. experimental evidence) this would really strengthen the paper.

We completely share the ambition to uncover functional properties of identified axolotl cells showing AER features. However, there are several significant technical limitations with the axolotl which would prevent us from obtaining clear data, and we think functional studies could be the focus of future work. Below, we detail the limitations we have, and our reasoning:

1. To address functionality, we considered classic manual AER removal experiments. However, first, axolotl AER-like cells do not appear to form a ridge structure that can be easily removed. Second, when the axolotl skin or even the entire limb bud is removed, the animal is able to regenerate the lost part (as in other regenerative species, e.g., PMID: 9441681, PMID: 20718005, PMID: 22485136), making it impossible to observe the phenotype of AER removal.
2. To test the functional equivalency, we then considered grafting AER-like cells from axolotls to other animals like mice and chickens. However, there are several problems with this approach. Firstly, the living temperatures of these animals are widely different. Thus, grafts most likely won't survive or they would exhibit abnormal behavior. Secondly, immune system incompatibility may cause host-rejection problems. Thirdly, in general, grafting experiments are not controllable and may involve additional injury responses. As manually isolating AER-like cells from axolotls cannot guarantee that the grafts would only contain AER-like cells, we would require a cell sorting strategy (e.g. based on fluorescent transgene and FACS) to produce cleaner grafts, which may take considerable time and animal usage yet would not be able to overcome the grafting-associated problems.
3. We also considered doing gene-basis functional work i.e. performing knockdown of *Vwa2* or other genes specifically expressed in axolotl AER-like cells. However, the mouse literature already demonstrated that most AER genes are functionally redundant and require double (and even triple knock-outs), which is tremendously challenging to establish with the axolotl as the reproductive cycle of axolotls is around 1-1.5 years. **But most critically, such gene-specific investigations would show the function of the gene, but not the cell type, which is the focus of our work.**
4. As the reviewer would appreciate, we consider the gold standard for cell type assessment is to perform cell type ablation experiments using genetic cell ablation (nitroreductase/metronidazole (NTR/Mtz) or diphtheria toxin A (DTA) systems). We (Sandoval-Guzmán Lab) have extensive experience working with the axolotl, and such methodologies are currently not available for axolotls, although our colleagues from multiple labs have tried them. In fact, we are aware that there are several groups which are currently working on developing cell type ablation systems in the axolotls.

Due to these reasons, functional assessment of the AER-like cells will require significant technological advances. **Nonetheless, critically, even if these cells are not functional (which seems extremely unlikely given their distinct gene expression profile), our conclusion would remain the same that axolotl limb buds contain cells with AER transcriptional program that are located at a similar, but not identical, spatial organization.** To emphasize this point, we have added additional discussion.

Revised manuscript in line 230:

“To gain further insights, future studies employing genetic cell ablation methods, which are currently unavailable in axolotls, could elucidate the functional role of the identified axolotl AER-like cells during limb development and whether they interact with the underlying mesenchyme as in other species. Regardless, our comprehensive evaluation provides compelling evidence that axolotl limb buds harbor cells with AER-like features, demonstrating both significant similarities in transcriptional programs and a substantial, albeit not identical, spatial organization.”

Reviewer: Mekayla Storer

Reviewer #2 (Remarks to the Author):

- What are the noteworthy results?

The present study has integrated and analyzed several previously generated vertebrate limb development/regeneration single cell transcriptomics datasets and have 3 major conclusions:

- 1) Axolotls possess Apical Ectodermal Ridge (AER) like cells during limb development.
- 2) Apical Ectodermal Cap (AEC) formed during axolotl limb regeneration and AER are distinct.
- 3) Axolotl mesoderm expresses AER factors.

- Will the work be of significance to the field and related fields? How does it compare to the established literature? If the work is not original, please provide relevant references.

Limb development studies have mostly been restricted to amniotes like chicken and mice that do not have the capacity to regenerate their limbs. Studies involving naturally occurring mutants, tissue excisions/transplantations and genetic ablations have identified the Apical Ectodermal Ridge or AER as one of the important signaling centers that triggers proximal-distal limb outgrowth. Fgfs like Fgf 4, 8, 9 and 17 are hallmarks of the AER and are key drivers of limb outgrowth. The AER-Fgfs also form a feedback loop with mesenchymal Shh and Grem1 to trigger limb outgrowth.

Salamander species like axolotls and newts are popular vertebrate models capable of limb regeneration. The question whether limb regeneration is a recapitulation of limb development is a long standing one. While there have been sporadic studies on limb development in salamanders, these have only explored a handful of genes and their functions. This study uses previously generated datasets to present an elaborate panel of markers for various tissue types involved during vertebrate limb development and specifically focuses on the ectodermal subset, AER cells to conclude that axolotls possess an AER. While this study is a good comparative analysis of various vertebrate models relevant to limb literature and for the first time uses spatial transcriptomics approach in axolotl limb regeneration paradigm, the first conclusion is overstated and needs substantive validations.

We thank the reviewer for the comments, which gave us an opportunity to highlight the technical strengths and limitations of our approach in our original submission.

In our study, we have undertaken a comprehensive and systematic evaluation by examining single cells from developing limbs in five different species, including a direct comparison between humans and axolotls, using sequencing-based methods and semi-quantitative HCR-based imaging. These approaches represent a significant advancement over previous studies that have primarily relied on morphological assessment or colorimetric in situ hybridization-based techniques.

We fully agree with the reviewer that the lack of functional validation prevents us from making the conclusive statement that "axolotls possess an AER". This awareness guided us to be cautious and precise in our interpretations and language throughout the manuscript. We stated that we have identified a distinct cell population in axolotl developing limbs that exhibits transcriptional features associated with the AER (manuscript line 64 and 74-76) and shows similar, but not identical, spatial organization. These conclusions are supported by our results and represent an accurate characterization of our findings. Therefore, we do not think our conclusion is an overstatement. Nonetheless, we agree with the reviewer that limitations on the functional properties of these cells should have been discussed. We now included an additional discussion on this point to clarify the validity of our conclusions.

The absence of canonical AER-Fgfs in the "axolotl-AER" and lack of any functional testing works against the claim.

The reviewer makes two different points related to how our work goes against our claim.

Regarding the comment of "The absence of canonical AER-Fgfs in the "axolotl-AER", we interpret this comment in two ways. Thus, we provide two responses.

- (1) If the reviewer means that our identified axolotl AER cells do not have AER-FGFs (*Fgf4*, *Fgf8*, *Fgf9*, and *Fgf17*), thus the identification of axolotl AER-like cells is not valid, correspondingly our response is as follows. We agree that the axolotl AER-like cells lack AER-FGFs. This observation is supported by the literature from multiple labs (PMID: 31538936, PMID: 35587651, PMID: 35531102). The lack of AER-FGFs is also one of the main reasons the literature indicated AER is absent in axolotls. Here, our work investigated the whole transcriptome unbiasedly (1000s of genes, not a selected few) to identify a cell population in axolotls showing AER characteristics. Therefore, we do not think our findings are contradictory to our claims. Instead, it **emphasizes the limitations of examinations based on a handful of genes and the critical need for unbiased transcriptome-wide studies for comparative studies**.
- (2) If the above statement considers any AER must have AER-FGFs which were identified in mice, our systematic analysis already underscores differences in FGF expression patterns across species (Supplementary Fig. 8a), indicating the flexibility in this cell state to guide limb morphogenesis. In fact, this flexibility has already been indicated by early studies showing chicken-specific *Fgf2* expression in AER (PMID: 8136521, PMID: 8200474, PMID: 7626791) that is not seen in mice (PMID: 11786528, PMID: 7626791). Thus, these results highlight that the AER program is not centered on the FGF ligands. There are many mutants established by independent studies where genes other than FGFs were deleted from AER, showing various defects in limb development (PMID: 12000792, PMID: 25166858, PMID: 19210962, PMID: 17661740).

Nonetheless, we are thankful for the reviewer's comment which encouraged us to further explore the flexibility in the AER program that we did not emphasize in our initial submission. A subset of axolotl AER-like cells have expression of *Fgf7* and *Fgf18*, instead of the canonical AER-FGFs (Supplementary Fig. 8a). We performed HCR staining for these genes which not only confirmed their expression in the axolotl AER-like cells, but also showed more mouse AER-like only dorsal-ventral boundary expression profile. We now included these results as Supplementary Fig 12 alongside a discussion on this critical point.

AER-specific Fgf expression in developing axolotl limbs.

Max-projection confocal image of axolotl Stage 53 hindlimb buds stained for *Fgf7*, and *Fgf18* mRNA via HCR. Gray, Hoechst; Cyan, *Fgf7*; Magenta, *Fgf18* mRNA; Scale bars: (Top) 100 μm and (Bottom) 20 μm .

Revised manuscript in line 127:

*“To explore species-specific features, we focused on the expressions of FGF ligands in the axolotl AER-like cells as they are well associated with the AER functions. We found that the FGF ligand expression profile showed variability across species, even between humans and mice (Supplementary Fig. 8). Specifically, scRNA-seq suggested that a subset of axolotl AER-like cells show specific expression of *Fgf7*, *Fgf16*, and *Fgf18*, although they did not express mouse AER-FGFs (*Fgf4*, *Fgf8*, *Fgf9*, *Fgf17*) (Supplementary Figs. 8). Strikingly, when we performed HCR against *Fgf7* and *Fgf18*, we found that they are dominantly expressed at the dorsal-ventral boundary of the distal limb bud ectoderm (Supplementary Fig 12), similar to the mouse AER FGFs. Taken together, these results indicate a diversification of FGF ligands in the axolotl ectoderm.”*

Revised manuscript in line 225:

*“Our results pave the way for a better understanding of the evolution of limb morphogenesis by highlighting the presence of transcriptionally highly similar populations, albeit showing distinct gene expression. Notably, the axolotl AER cells do not appear to express AER-FGFs (*Fgf4*, *Fgf8*, *Fgf9*, *Fgf17*), aligning with previous reports^{6,7}. Instead, they express different FGFs (e.g., *Fgf7* and *Fgf18*), stressing the utility of high-throughput and unbiased methodologies for cross-species comparisons.”*

Regarding the comment on functionality, which is also raised by the Rev1, we provide our response to this very valid point below.

We completely share the ambition to uncover functional properties of identified axolotl cells showing AER features. However, there are several significant technical limitations with the axolotl which would prevent us from obtaining clear data, and we think functional studies could be the focus of future work. Below, we detail the limitations we have, and our reasoning:

1. To address functionality, we considered classic manual AER removal experiments. However, first, axolotl AER-like cells do not appear to form a ridge structure that can be easily removed. Second, when the axolotl skin or even the entire limb bud is removed, the animal is able to regenerate the lost part (as in other regenerative species, e.g. PMID: 9441681), making it impossible to observe the phenotype of AER removal.
2. To test the functional equivalency, we then considered grafting AER-like cells from axolotls to other animals like mice and chickens. However, there are several problems with this approach. Firstly, the

living temperatures of these animals are widely different. Thus, grafts most likely won't survive or they would exhibit abnormal behavior. Secondly, immune system incompatibility may cause host-rejection problems. Thirdly, in general, grafting experiments are not controllable and may involve additional injury responses. As manually isolating AER-like cells from axolotls cannot guarantee that the grafts would only contain AER-like cells, we would require a cell sorting strategy (e.g. based on fluorescent transgene and FACS) to produce cleaner grafts, which may take considerable time and animal usage yet would not be able to overcome the grafting-associated problems.

3. We also considered doing gene-basis functional work i.e. performing knockdown of *Vwa2* or other genes specifically expressed in axolotl AER-like cells. However, the mouse literature already demonstrated that most AER genes are functionally redundant and require double (and even triple knock-outs), which is tremendously challenging to establish with the axolotl as the reproductive cycle of axolotls is around 1-1.5 years. **But most critically, such gene-specific investigations would show the function of the gene, but not the cell type, which is the focus of our work.**

As the reviewer would appreciate, we consider the gold standard for cell type assessment is to perform cell type ablation experiments using genetic cell ablation (nitroreductase/metronidazole (NTR/Mtz) or diphtheria toxin A (DTA) systems). We (Sandoval-Guzmán Lab) have extensive experience working with the axolotl, and such methodologies are currently not available for axolotls, although our colleagues from multiple labs have tried them. In fact, we are aware that there are several groups which are currently working on developing cell type ablation systems in the axolotls.

Due to these reasons, functional assessment of the AER-like cells will require significant technological advances. **Nonetheless, critically, even if these cells are not functional (which seems extremely unlikely given their distinct gene expression profile), our conclusion would remain the same that axolotl limb buds contain cells with AER transcriptional program that are located at a similar, but not identical, spatial organization.** To emphasize this point, we have added additional discussion.

Revised manuscript in line 230:

"To gain further insights, future studies employing genetic cell ablation methods, which are currently unavailable in axolotls, could elucidate the functional role of the identified axolotl AER-like cells during limb development and whether they interact with the underlying mesenchyme as in other species. Regardless, our comprehensive evaluation provides compelling evidence that axolotl limb buds harbor cells with AER-like features, demonstrating both significant similarities in transcriptional programs and a substantial, albeit not identical, spatial organization.."

The authors also need to explore the placement of the feedback loop and interaction between the ectoderm and the mesenchyme during limb outgrowth in axolotls.

The suggestion to investigate the interactions between the ectoderm and mesenchyme, particularly in the context of an AER without canonical FGFs, is indeed an interesting and valuable direction for future research. The literature has extensively examined mesenchymal-epithelial interactions mediated by canonical AER-FGFs. In our current work, we have resolved the conflicting statements and assumptions regarding the cells with AER features in axolotls and to what extent they are reused during limb regeneration by employing comprehensive and quantitative methodologies. These significant advances allow future studies to explore how limb ectoderm lacking canonical FGFs operates with mesodermal cells exhibiting AER features and presents a novel and intriguing perspective on limb morphogenesis.

While we acknowledge that investigating the interactions between the ectoderm and mesenchyme would be an exciting avenue of exploration, we think it is beyond the scope of this current work. However, this is indeed a critical exciting next step to emphasize, we now included it in our discussion.

Revised manuscript in line 230:

"To gain further insights, future studies employing genetic cell ablation methods, which are currently unavailable in axolotls, could elucidate the functional role of the identified axolotl AER-like cells during limb development and whether they interact with the underlying mesenchyme as in other species."

The finding that AEC and AER are distinct is novel and partially answers the paradox whether limb regeneration is a recapitulation of limb development from the ectodermal cell perspective alone. The

finding that the axolotl mesoderm possesses AER factors has been previously reported in axolotl limb development (Purushothaman et al., 2019; Glotzer et al., 2022) and axolotl limb regeneration (Nacu et al., 2016) albeit based on a handful of markers, and this study further extends the panel of genes with few validations based on spatial transcriptomics.

We appreciate the reviewer's comments on our work. Indeed, the comparison between limb regeneration and development has long been a subject of debate. Our previous research with *Xenopus laevis* tadpoles has shown that regeneration in this species recapitulates development for ectodermal cells (PMID: 34105722). However, our current findings in axolotls suggest that the situation is different which challenges our assumptions about the commonalities of all limb regeneration scenarios and provides evidence for the involvement of different cell states in limb regeneration paradigms.

• Does the work support the conclusions and claims, or is additional evidence needed?

The conclusions and claims here are based on single cell transcriptomics analyses with validation of few genes using HCR FISH and Visium.

For axolotl limb development, HCR FISH validation is only done for 2 genes *Dr999* and *Vwa2* that do not have comparative expression profiles in mouse and chicken limb literature. HCR FISH for these genes on mouse or chicken can partially support the claim.

Thanks for this suggestion. To address the reviewer's point, we first reviewed our scRNA-Seq dataset and found that *Vwa2* (unlike *Dr999-Pmt21178*, which has no known orthologs with other species analyzed) is expressed at different levels in mouse, frog, and human AER cells (see below). Similar to the axolotl, the expression of *Vwa2* in the AER is higher than in the basal epidermis in these species. We did not detect *Vwa2* expression in any cells in chicken which could be due to the low sequencing quality of this dataset, as the reviewer also noted above. Then, we performed HCR for *Fgf8* (to label AER) and *Vwa2* on mouse and chicken limbs. In alignment with sequencing results, mouse *Vwa2* is expressed in the distal site of the limb bud, labelling both AER and the surrounding basal ectoderm. Similarly, our staining results for chicken *Vwa2* also showed its expression in the AER, although it could also be seen in other ectodermal cells. Altogether, these staining results further confirm the validity of our cross-species comparison and the conclusion about the axolotl limb buds containing cells with the AER transcriptional program.

Vwa2 expression in humans, mice, chickens, and frogs.

d) Violin plots showing *Vwa2* expression across all cell types in indicated species and developmental stages based on the analyzed scRNA-Seq datasets.

- e) Max-projection confocal image of mouse E10.5 hindlimb limb buds stained for *Fgf8*, and *Vwa2* mRNA via HCR. Gray, Hoechst; Magenta, *Fgf8* mRNA; Cyan, *Vwa2* mRNA. Scale bars in all images: 100 μ m
- f) Max-projection confocal image of chicken HH22 hindlimb buds stained for *Fgf8*, and *Vwa2* mRNA via HCR. Gray, Hoechst; Magenta, *Fgf8* mRNA; Cyan, *Vwa2* mRNA. Scale bars: (Top) 250 μ m and (Bottom) 25 μ m.

Revised manuscript in line 119:

“Since our results revealed *Vwa2* and *Dr999-Pmt21178* as markers for the AER-like cells in axolotls, we then examined their expression in the AER in other species. Unlike *Dr999-Pmt21178* which does not have orthologs in other analyzed species, *Vwa2* are expressed in different levels in the AER of humans, mice, and frogs in scRNA-seq data (Supplementary Fig 11a). Meanwhile, we did not detect any *Vwa2*-expressing cells in the chicken dataset, which may be due to its low sequencing depth (Supplementary Figs. 1 and 11a). Indeed, performing HCR on developing chicken or mouse limb buds confirmed the *Vwa2* expression in mouse and chicken AERs (Supplementary Figs. 11b and c). Thus, these results indicate that employing cross-species comparisons can unveil novel cell type markers.”

Additionally, as also suggested by the Rev 1, we now performed AER marker *Msx2* staining in the axolotl, which revealed co-localized expression with *Dr999-Pmt21178*. Altogether, these staining results further confirm the validity of our cross-species comparison and the conclusion about the axolotl limb buds containing cells with the AER transcriptional program. We added these results as Supplementary Fig. 10a-b and revised the text accordingly.

***Dr999-Pmt21178*+ Axolotl AER-like cells express *Msx2* and not all *Epcam*+ skin cells express *Dr999-Pmt21178*.**

(Left) Max-projection confocal image of axolotl Stage 53 hindlimb buds stained for *Dr999-Pmt21178* (please note the gene name is labelled *Dr999* in the figure), and *Msx2* mRNA via HCR. Scale bar: 100 μ m. Red arrows show skin cells, and the green arrows show non-skin mesodermal cells to highlight *Msx2* staining profile.

(Right) Close up images of *Dr999* and *Msx2* double positive skin cells. Scale bar: 10 μ m. (b) Expression of *Msx2* in stage 50 (top), 52 (middle) and 54 (bottom) axolotl limb buds based on the scRNA-Seq datasets. Cell type labels are adopted from Supplementary Fig. 2e.

Revised manuscript in line 114:

“We then showed that *Dr999-Pmt21178* positive cells also co-express a known AER marker *Msx2* (Supplementary Fig. 10a-b).”

Additionally, *Dr999* expression at stage 50 and 52 (Supplementary Fig.9 c) is not convincing as it does not show the classic AER like domain and is present throughout the ectoderm.

Thanks to both reviewers, we realized that our description of axolotl AER spatial organization and expression profile of *Dr999-Pmt21178* and *Vwa2* were not clear enough. In our original manuscript, we stated (line 97) that the localization of *Dr999-Pmt21178* is not the same as the classic AER domain but “more scattered compared to AER localization in other species”. Indeed, the spatial difference might be also one of the reasons why axolotls are considered to be without an AER. To make our findings clear on this point, we now revised the text as below.

Revised manuscript in line 113:

“...resembling, but not identical to, the spatial organization of AER in other species^{1,2}.”

To address the concern about *Dr999-Pmt21178* being expressed throughout the ectoderm, we performed co-staining of *Epcam*, a pan-ectoderm marker, which shows that not all *Epcam* positive cells are *Dr999-Pmt21178* positive (see below). We now added these results as Supplementary Fig. 10d, and edited text accordingly to clarify these very important details.

Max-projection confocal image of axolotl Stage 53 hindlimb buds stained for *Dr999-Pmt21178*, and *Epcam* mRNA via HCR. Gray, Hoechst; Magenta, *Dr999-Pmt21178* mRNA; Cyan, *Epcam* mRNA. Scale bar: 200 μ m.

Revised manuscript in line 115:

“Moreover, we found that *Dr999-Pmt21178* expression colocalized with the pan-ectoderm marker *Epcam* only in the distal limb bud ectoderm (Supplementary Fig 10c), suggesting that the axolotl AER-like cells are distinct from non-AER ectoderm.”

Previous studies (Purushothaman et al., 2019, Ghosh et al., 2008) show mesenchymal expression for genes like *Msx2* and *Wnt5a*, while single cell transcriptomics data here shows ectodermal expression.

The expression profiles for these genes have been confusing in the literature. As the reviewer indicates, Purushothaman et al. suggest that *Msx2* is in the mesoderm. However, there is another study suggesting *Msx2* is in both mesoderm and ectoderm, which is consistent with our scRNA-seq datasets (see below). (PMID: 9846383). Meanwhile, Ghosh et al and a more recent paper showed that *Wnt5a* is both in ectoderm and mesoderm (PMID: 18336582, PMID: 35531102).

The conflicting expression patterns of *Msx2* (and other markers) in the literature bring us to the critical point that single gene examinations may be misleading. Staining will be influenced by the use of probe specificity and may produce different outcomes with changes in gene annotations. **That is why our work infers potential cell types by assaying the whole transcriptome profiles rather than limiting to a few genes**, which we believe could provide a more accurate and comprehensive understanding of cell types and their characteristics. We hope the reviewer also shares this perspective with us.

To clarify the expression of *Msx2* in axolotl limb buds, we performed HCR and confirmed its expression in both mesoderm and ectoderm, as well as its co-expression with *Dr999-Pmt21178*. We added these results as Supplementary Fig. 10a-c and revised the text accordingly.

***Dr999-Pmt21178+* Axolotl AER-like cells express *Msx2* and not all *Epcam+* skin cells express *Dr999-Pmt21178*.** (Left) Max-projection confocal image of axolotl Stage 53 hindlimb buds stained for *Dr999-Pmt21178* (please note the gene name is labelled *Dr999* in the figure), and *Msx2* mRNA via HCR. Scale bar: 100 μ m. Red arrows show skin cells, and the green arrows show non-skin mesodermal cells to highlight *Msx2* staining profile. (Right) Close up images of *Dr999* and *Msx2* double positive skin cells. Scale bar: 10 μ m. (b) Expression of *Msx2* in stage 50 (top), 52 (middle) and 54 (bottom) axolotl limb buds based on the scRNA-Seq datasets. Cell type labels are adopted from Supplementary Fig. 2e.

Revised manuscript in line 114:

*“We then showed that *Dr999-Pmt21178* positive cells also co-express a known AER marker *Msx2* (Supplementary Fig. 10a-b).”*

The authors must depict axolotl specific, all cell type UMAP plots and violin plots (similar to Supplementary Fig.9 b) for all the genes in the AER specific list and provide HCR FISH/Visium validations for highly expressed genes to confirm exclusive expression in the AER.

We thank the reviewer for this comment which made us realize the strength of our methodology could have been conveyed more clearly.

Here we used scRNA-Seq to identify **marker genes** for the axolotl AER, which is a widely-used approach successfully identifying differentially expressed genes representative of cell types and their associated whole transcriptomes. Based on this, we validated the two scRNA-seq-derived markers for the AER-like populations, *Dr999-Pmt21178* and *Vwa2* (Supplementary Fig. 9c) as these two genes have the highest expression and specificity to AER (Supplementary Fig. 9a, see below). Therefore, we can confidently consider cells marked by *Dr999-Pmt21178* and *Vwa2* in the HCR images effectively share a high degree of gene expression profile similarity with the AER-like population identified by scRNA-seq. As this is the case, performing more staining (all axolotl AER markers, 14 genes as suggested by the reviewer) would be redundant.

The heatmap (Supplementary Fig. 9a) was initially chosen to present the axolotl marker genes in a compact format, allowing for the visualization of gene expression patterns across different cell populations. As the reviewer suggested, we now plotted UMAPs and violin plots for these genes, please see below. We consider adding individual UMAPs and violin plots to convey the same message as the heatmap. Therefore, we prefer to not include them in the manuscript, but we could if the reviewer wishes.

Axolotl ST50

UMAP plots (top panel) and corresponding violin plots (bottom panel) showing expression levels of the putative axolotl AER markers in stage 50 axolotl limb buds. AER-like cells are labelled in text in the first UMAP plot on the upper left. For full annotation, see Supplementary Fig. 2e.

Axolotl ST52

UMAP plots (top panel) and corresponding violin plots (bottom panel) showing expression levels of the putative axolotl AER markers in stage 52 axolotl limb buds. AER-like cells are labelled in text in the first UMAP plot on the upper left. For full annotation, see Supplementary Fig. 2e.

Axolotl ST54

UMAP plots (top panel) and corresponding violin plots (bottom panel) showing expression levels of the putative axolotl AER markers in stage 54 axolotl limb buds. AER-like cells are labelled in text in the first UMAP plot on the upper left. For full annotation, see Supplementary Fig. 2e.

• Are there any flaws in the data analysis, interpretation and conclusions? Do these prohibit publication or require revision?

The methodology used for single cell transcriptomics analysis, HCR and Visium is sound however the interpretations and conclusions regarding axolotl AER is overstated for reasons mentioned above. The paper

requires major revisions to support the claim that axolotls possess an AER, including validation of the expression of AER markers and functional analysis of axolotl ectoderm and its association with the underlying mesenchyme.

- **Is the methodology sound? Does the work meet the expected standards in your field?**

The methodology for single cell transcriptomics analysis, HCR and Visium is sound and meets the expected standards in the field. One suggestion for imaging whole mount limbs is to also have the lateral view images to verify if the Dr999 and Vwa2 are expressed at the dorsal-ventral border of the limb buds/digits.

- **Is there enough detail provided in the methods for the work to be reproduced?**

The methodology is elaborate enough and can be reproduced.

We thank the reviewer for their positive remarks in the above three sections. We have made efforts to address the reviewer's comments and suggestions by providing additional validation staining, clarifying our language, and emphasizing the strengths and limitations of our approach. As discussed above, we agree with the reviewer that while functional characterization of the identified axolotl cells is an important area for future investigation, our current conclusions are supported by the data we have presented. The identification of these AER-like cells in axolotls, even if their function differs or remains unknown, lays the groundwork for further exploration into their functionality and their interactions with the mesenchyme.

Reviewer #1 (Remarks to the Author):

I applaud the authors for their efforts in providing sufficient evidence through experimental data, additional clarification of methodology or explanation to satisfy all of my concerns. While the authors were unable to provide functional validation that this mesenchymal population of cells function in a similar manner to the AER, this does appear to be a limitation of the model organism and is consistent with the current state of the field. Despite this, the manuscript clearly represents a clear advancement to our current knowledge and I am wholly supportive of its publication.

Reviewer #2 (Remarks to the Author):

> Previous studies have identified AER based on the morphology, markers including Fgfs and functional role in proximal-distal outgrowth. The present study does not associate any of the above characteristics with AER and puts forward a new term "AER-like cells" in axolotls based on single cell transcriptomics analysis and HCR FISH to prove that this is a relatively expanded domain.

> Cross species comparison using single cell datasets and use of Visium in axolotls are novel aspects of this study, but validations with HCR FISH are not convincing. HCR FISH images for Fgf7, Fgf18, Vwa2 and Dr999 have low signal:noise ratio and have signals all over the ectoderm (supplementary Fig 9 and 12). Dr999 signals overlap with the pan ectoderm marker Epcam (supplementary Fig 10c).

> It is worth noting that the authors tried HCR FISH for Vwa2 and Dr999 on mice and chicken limb bud, but all have signal in the basal ectoderm and cannot be considered as an exclusive AER marker. The authors have Fgf8 FISH alongside but also state in the response that presence of AER cannot be centered around Fgfs.

> The absence of functional testing in axolotl model is reasonable and is a long term study.

> This manuscript needs clear HCR FISH images that can substantiate the single cell transcriptomics findings.

We thank the reviewer 1 for their positive comments and support for publication, and the editor and reviewer 2 for their comments. We have now addressed all the new points raised by the reviewer 2 below by adding clarifications, enlarged images, and quantifications as kindly suggested by the editor. The reviewers' comments are in bold blue, and our response is in black.

Reviewer #1 (Remarks to the Author):

I applaud the authors for their efforts in providing sufficient evidence through experimental data, additional clarification of methodology or explanation to satisfy all of my concerns. While the authors were unable to provide functional validation that this mesenchymal population of cells function in a similar manner to the AER, this does appear to be a limitation of the model organism and is consistent with the current state of the field. Despite this, the manuscript clearly represents a clear advancement to our current knowledge and I am wholly supportive of its publication.

We thank the reviewer for their positive remarks and their endorsement for the publication of our manuscript.

Reviewer #2 (Remarks to the Author):

Previous studies have identified AER based on the morphology, markers including Fgfs and functional role in proximal-distal outgrowth. The present study does not associate any of the above characteristics with AER and puts forward a new term "AER-like cells" in axolotls based on single cell transcriptomics analysis and HCR FISH to prove that this is a relatively expanded domain.

We thank the reviewer for their comments. Indeed, the AER is characterized on the basis of gene expression, functionality, and morphology, as well as its unique spatial organization. However, we respectfully disagree with the statement that our work does not associate any of the AER characteristics with the cells we identified.

Specifically, our study provides clear data that there are skin cells in axolotls that share the AER transcriptional signature based on the transcriptome similarity, not just a handful of genes. Such a transcriptome-wide comparison approach alone has been widely used in many single-cell transcriptomics studies to detect cell types with high confidence. However, in our work, we extended our computational observation to also reveal spatially and morphologically unique aspects of AER-like cells in the axolotl. Based on our computational analysis, we identified highly expressed genes in the axolotl cells to reveal the location of these cells, which are similar, but not identical, to the AER in other species. In addition, we found that these cells have a flattened cell shape, which is different from the AER in other species. We have outlined our rationale for the lack of functional assays in our previous response letter, which is supported by both reviewers. We believe that such axolotl-specific differences and limitations may have contributed to the failure to identify this population previously, making it elusive. Overall, we provide compelling evidence for 2/3 of the criteria mentioned by the reviewer (3/4 based on our criteria, including unique spatial organization), making our study a landmark for further investigation of axolotl AER-like cells in the future.

We believe that labeling this population as AER or AER-like is a semantic issue. Based on our results, we have very carefully described what we consider to be the similarity of the identified axolotl cells to AER cells in other species in both quantitative measures (using scRNA-Seq) and qualitative assessments (HCR spatial organization and cell morphology results), so that the reader can determine the extent to which these cells resemble AER or whether they should be called AER-like. We hope that the reviewer shares our view on this point.

Cross species comparison using single cell datasets and use of Visium in axolotls are novel aspects of this study, but validations with HCR FISH are not convincing. HCR FISH images for Fgf7, Fgf18, Vwa2 and Dr999 have low signal:noise ratio and have signals all over the ectoderm (supplementary Fig 9 and 12). Dr999 signals overlap with the pan ectoderm marker Epcam (supplementary Fig 10c).

We thank the reviewer for their positive remarks, and are sorry to hear that our HCR images were not considered convincing to this reviewer. In our paper, we presented state-of-the-art HCR FISH data either as whole-mount images or as zoomed-in single-cell confocal optical sections. We take this opportunity to stress why HCR is an adequate method for our purposes, which is becoming more widely used in the recent literature, especially as a validation approach for scRNA-Seq.

HCR is a semi-quantitative method, but it also has its limitations. Highly or lowly expressed genes (as assessed by scRNA-Seq) have an expected higher or lower HCR signal:noise ratios, respectively. HCR is not similar to the traditionally used colorimetric in situ hybridization, which amplifies the signal for even lowly expressed genes. On this basis, the images we present are of high quality and in line with the standards of the field. Moreover, all images we present are consistent with our scRNA-Seq results (considering the quantitative discriminative value of scRNA-Seq and its complementarity with HCR), validating the expression profile while showing novel (*Dr999*, *Vwa2*, *Fgf7*, *Fgf18*) and expected (*Epcam*, *Msx2*) spatial organization for different genes in different species. Lastly, we would also like to point out that our labs (Aztekin and Sandoval-Guzman labs) have extensive experience with HCR, as we have been using this method routinely for the last four years, testing more than 65 different probes in five model systems. Thus, our experience is sufficient to discriminate when HCR does not work or produces low signal-to-noise ratios that we would not consider convincing.

In our original submission we had images for *Dr999* and *Vwa2* and the reviewer did not raise any specific concerns. In our revision, we have included axolotl *Fgf7*, *Fgf18*, *Msx2* and *Epcam*, mouse *Vwa2* and chicken *Vwa2* HCR images. All of these gene expression profiles are consistent with the scRNA-Seq datasets and further support our findings regarding the presence of an AER-like population in the axolotl. The reviewer is concerned about 4 out of 8 (*Fgf7*, *Fgf18*, *Vwa2*, and *Dr999*) genes (although the species of *Vwa2* is not specified, and *Dr999* and *Vwa2* were not previously a concern), and cites two problems with them: (1) low signal:noise ratio, and (2) signals throughout the ectoderm.

For the reviewer, we provide zoomed images for some of the concerned images (we used the same format and LUTs to keep the data consistent and since these images are criticized). Our zoomed images clearly show specific expression profiles while also showing that the expression profiles are not seen throughout the ectoderm (please see below), in contrast to the reviewer's concern. We also add quantifications, as suggested by the editor, to show spatially enriched areas which demonstrated none of the criticized signals are seen throughout the ectoderm (please see below). We believe that adding these additional zoomed images and quantifications would be redundant to the results presented. Therefore, we prefer not to include them in the main text (we are happy with their inclusion in the peer review document), but we could do so if the reviewer wishes.

Axolotl *Dr999* and *Vwa2*

The scRNA-Seq results show that *Dr999* has a high expression profile (normalized UMI around 4), while *Vwa2* has a lower expression (normalized UMI around 3). In alignment with this, we see that *Dr999* has a high signal:noise ratio and *Vwa2* has a lower but clear signal:noise ratio (based on the number of punctas in each cell).

Regarding the comment about *Dr999* is seen throughout the ectoderm, please see below *Epcam/Dr99* section. Likewise, for the comment about *Vwa2* is seen throughout the ectoderm, in our original submission and during revision, we stated that *Vwa2* is also expressed in some but not all basal ectoderm, according to our scRNA-Seq result. We then also confirmed this finding using HCR (discussed in our original submission, in detail in the first round of revisions, and also please see below).

For the reviewer, we now provide zoomed images for *Dr999* (both here and below in the *Epcam* section) and *Vwa2*. The images below zoomed to two different regions show that high *Dr999* and *Vwa2* expressing cells are mainly located at the distal end (number 1). We also show lower *Vwa2* expressing cells or no *Vwa2* expressing cells from the proximal parts of the limb bud. *Dr999* expression is already not seen in proximal regions. These

images further demonstrate the quantitative nature of HCR and its alignment with scRNA-Seq.

***Dr999-Pmt21178* and *Vwa2* expression in axolotl developing limb buds.**

- a) Max-projection confocal images of Stage 46 axolotl forelimbs correspond to ST50 hindlimb scRNA-Seq data, stained for *Dr999-Pmt21178* and *Vwa2* mRNA via HCR. In the merged figures green corresponds to *Dr999-Pmt21178* mRNA (referred to as *Dr999* in the figure), red corresponds to *Vwa2* mRNA, and blue corresponds to Hoechst. Scale bar 100 μ m. Please note that this image is the same as Supplementary Fig. 9c.
- b-c) Enlarged views of the regions in (a) indicated by yellow box number 1 (b) and number 2 (c). Scale bar 20 μ m.

Mouse and chicken *Vwa2*

Both mouse and chicken *Vwa2* expressions are low based on the analyzed scRNA-Seq datasets (Supplementary Fig. 11a). Thus, it is expected to see a lower signal:noise ratio.

For the reviewer, we now provide zoomed views showing areas with varying *Vwa2* expression, stressing the specificity of the used probes. These zoomed images again show us that *Vwa2* has a lower but clear signal:noise ratio.

Regarding the comment that the signal is throughout the ectoderm, we have already discussed *Vwa2* being also seen in both basal ectoderm and AER both in original submission, during the first revision, and above. Altogether, these results show that *Vwa2* expression is similar across mice, chicken, and the axolotl.

***Vwa2* expression in mice and chickens developing limb buds.**

- a) (Left) Max-projection confocal image of mouse E10.5 hindlimb limb buds stained for *Fgf8*, and *Vwa2* mRNA via HCR. Gray, Hoechst; Magenta, *Fgf8* mRNA; Cyan, *Vwa2* mRNA. (Right) Enlarged views of the regions in (a) indicated by yellow box number 1 (top) and 2 (bottom). Scale bar: (Left) 100 μm and (Right) 10 μm . Please note that this image is the same as Supplementary Fig. 11b.
- b) Max-projection confocal image of chicken HH22 hindlimb buds stained for *Fgf8*, and *Vwa2* mRNA via HCR. Gray, Hoechst; Magenta, *Fgf8* mRNA; Cyan, *Vwa2* mRNA. (Right) Enlarged views of the regions in (a) indicated by yellow box number 1 (top), 2 (middle) and 3 (bottom). Scale bars: (Left) 100 μm and (Right) 10 μm . Please note that this image is the same as Supplementary Fig. 11c.

Axolotl *Fgf7* and *Fgf18*

We agree that *Fgf7* and *Fgf18* expressions do indeed have a lower signal to noise ratio. However, the signals are still clear and are enriched at the distal tip of limb buds (see below) and show a highly specific expression profile, revealing an intriguing spatial organization similar to AER in other species. Moreover, the analyzed axolotl scRNA-Seq indicates that some, but not all, axolotl AER-like cells show this gene expression (Supplementary Fig. 8a), which might be related to the spatial organization we see.

For the reviewer, we now provide zoomed areas to distinguish high or low *Fgf7* and *Fgf18* areas. We also quantified, as suggested by the editor, the *Fgf7* and *Fgf18* signal intensities using the Fiji plot profile for the selected area (Panel d, quantified area is labelled by the green box). The green arrow indicates the distance from proximal to distal. We could see a clear enrichment at the tip for *Fgf7*, which also shows a higher signal:noise ratio. However, in contrast to our qualitative evaluation based on zoomed images, we see that this is less pronounced for *Fgf18*. Particularly, we do see more background at the proximal limb buds, but this background is decreased towards the distal site and we see the signal is peaking at the distal limb bud. Critically, the *Fgf18* signal is clearly seen as discrete puncta at the distal limb bud tip, and this phenotype is not seen in any other place, including the whole limb ectoderm. Here we would like to stress that such a specific expression pattern cannot be explained by having low signal:noise. Overall, these results again suggest that

these genes are not expressed throughout the ectoderm at the levels seen in the distal tip.

AER-specific Fgf expression in developing axolotl limbs.

- Max-projection confocal image of axolotl Stage 53 hindlimb buds stained for *Fgf7* and *Fgf18* mRNA via HCR. Gray, Hoechst; Cyan, *Fgf7*; Magenta, *Fgf18* mRNA; Scale bars: 100 μm . Please note that this image is the same as Supplementary Fig. 12.
- Zoomed region of (a) indicated by yellow box number 1. Scale bars: 20 μm .
- Zoomed region of (a) indicated by yellow box number 2. Scale bars: 20 μm .
- The same image as (a) labelled with a green box to indicate the quantification area and an arrow to indicate the direction of quantification.
- Quantification of *Fgf7* signal intensity along the green arrow in (d).
- Quantification of *Fgf18* signal intensity along the green arrow in (d).

About Epcam and Dr999 staining

Regarding the comment "Dr999 signals overlap with the pan ectoderm marker Epcam (Supplementary Fig. 10c)". We are not sure if we understood the reviewer correctly.

Epcam is an ectodermal marker and is expected to be present in the AER and other skin cell types. Our maximum projection limb bud image directly shows this to be the case.

Specifically, we performed a *Dr999* and *Epcam* co-staining experiment in response to the reviewer's initial comment that *Dr999* was present throughout the ectoderm. This previously submitted co-staining clearly shows that this is not the case: *Dr999* is enriched at the distal tip, and there are regions where *Dr999* is absent but *Epcam* is expressed.

We now provide zoomed images of the same limb bud staining showing a high (number 1), low (number 2), or absent (number 3) *Dr999* expression profile. These 3 zoomed images are selected from distal to proximal areas - designated by numbers. We further quantified, as suggested by the editor, *Dr999* signal intensity using Fiji plot profile with the green arrow direction indicating distal-proximal distance (Panel b, the quantified area is labelled by the green box). This quantification again showed that the *Dr999* profile is not uniform throughout the ectoderm. We hope that our additional zoomed images and quantifications will resolve the concerns about the expression profile of *Dr999* and *Epcam*.

Not all *Epcam*+ skin cells express *Dr999-Pmt21178*.

- Max-projection confocal image of axolotl Stage 53 hindlimb buds stained for *Dr999-Pmt21178*, and *Epcam* mRNA via HCR. Gray, Hoechst; Magenta, *Dr999-Pmt21178* mRNA; Cyan, *Epcam* mRNA. Scale bar: 100 μ m for the whole limb bud, and 20 μ m for zoomed images. Please note that this image is the same as Supplementary Fig. 10c. Enlarged views of corresponding regions in (a) indicated by the numbered yellow boxes are shown side by side.
- Quantification of *Dr999* signal intensity in (a). The quantified area is indicated in the green box in the inserted image. Quantification was performed along the direction indicated by the green arrow.

It is worth noting that the authors tried HCR FISH for *Vwa2* and *Dr999* on mice and chicken limb bud, but all have signal in the basal ectoderm and cannot be considered as an exclusive AER marker. The authors have *Fgf8* FISH alongside but also state in the response that presence of AER cannot be centered around *Fgfs*.

We thank the reviewer who suggested that we check the expression of *Dr999* and *Vwa2* in other species to see if it is similar to that in the axolotl. In the axolotl, we found that high *Vwa2* expression is seen in the AER-like cells, but lower *Vwa2* expression could also be seen in other ectodermal cells (as discussed before both during our initial submission and more extensively during revision). On this basis, our results with mouse and chicken *Vwa2* staining showed highly similar expression pattern as in the axolotl results: *Vwa2* is expressed in the AER, and it is also seen in the basal ectoderm in chickens and mice.

However, we think there are several misunderstandings about our new experiments. Contrary to the reviewer's comment, we only performed HCR for *Vwa2* in mouse and chicken, but not for *Dr999*, as it has no orthologs in the species analyzed (which is already indicated in the revised manuscript).

Secondly, we never suggested that *Vwa2* was an exclusive AER marker, and we have emphasized this situation in both our original submission, and during the first revision. Specifically, we stated that *Vwa2* is also seen in the basal ectoderm based on scRNA-Seq and HCR images similar to the axolotl AER-like population (manuscript line 119). Our mice and chickens staining results showed that the *Vwa2* domain overlaps with the AER (marked by *Fgf8*), albeit with its broader expression (Supplementary Fig. 11b, c). This expression pattern is consistent with *Vwa2* in axolotl limb buds (Supplementary Fig. 9c).

As the reviewer also suggested in the previous revision, *Vwa2* showing a similar expression pattern in axolotls as in mice and chickens would further support the identification of the AER-like cells, and we thank the reviewer for this suggestion.

The authors have *Fgf8* FISH alongside but also state in the response that presence of AER cannot be centered around *Fgfs*.

We agree with the reviewer that *Fgf8* is indeed an important marker and has a highly AER-specific expression profile in mouse, chicken, human and frog datasets. However, our results highlight that despite the axolotl lacking an ectodermal *Fgf8*, there are still cells that share a highly significant transcriptome-wide similarity to the AER in other species, suggesting that the AER transcriptional program may not be centered around *Fgf8* when these 5 species are considered.

Our results show a transcriptome-wide similarity and scRNA-Seq-based predicted axolotl AER-like cell gene expressions, some of which are already provided in the original submission (*Dr999* and *Vwa2*) and this list has been significantly expanded (with *Fgf7*, *Fgf7*, *Fgf18*, *Epcam*, and *Msx2*) during the first revision thanks to reviewer comments (in total, *Vwa2*, *Dr99*, *Fgf7*, *Fgf18*, *Msx2*, *Epcam*), all of which are located in the expected spatial patterns of axolotl AER-like cells, as well as in agreement with scRNA-Seq results.

The absence of functional testing in axolotl model is reasonable and is a long term study.

We thank the reviewer for their understanding.

This manuscript needs clear HCR FISH images that can substantiate the single cell transcriptomics findings.

As mentioned above, our HCR FISH images are clear, of high quality, and within the standards of the field. To alleviate the reviewer's concerns about HCR quality, we have now included quantifications to this letter and zoomed in on images related to raised points, all of which support our previous conclusions based on qualitative assessments.